# Binding and sequestration of poison frog alkaloids by a plasma globulin

**Aurora Alvarez-Buylla[1]\*, Marie-Therese Fischer[1], Maria Dolores Moya Garzon[2,3,4], Alexandra E Rangel[5], Elicio E Tapia[6], Julia T Tanzo[2,3], H Tom Soh[5,7,8], Luis A Coloma[9], Jonathan Z Long[2,3,4,10], Lauren A O'Connell[1,3,11]\***

[1]Department of Biology, Stanford University, Stanford, United States; [2]Sarafan ChEM-H, Stanford University, Stanford, United States; [3]Wu Tsai Institute for Neuroscience, Stanford University, Stanford, United States; [4]Department of Pathology, Stanford University, Stanford, United States; [5]Wu Tsai Human Performance Alliance, Stanford University, Stanford, United States; [6]Department of Radiology, Stanford University, Stanford, United States; [7]Center for Taxonomy and Morphology, Leibniz Institute for the Analysis of Biodiversity Change, Hamburg, Germany; [8]Department of Electrical Engineering, Stanford University, Stanford, United States; [9]Chan Zuckerberg Biohub, San Francisco, United States; [10]Centro Jambatu de Investigación y Conservación de Anfibios, Fundación Jambatu, San Rafael, Ecuador; [11]Stanford Diabetes Research Center, Stanford University, Stanford, United States

\*For correspondence:
auroraab@stanford.edu (AA-B);
loconnel@stanford.edu (LAO'C)

**Competing interest:** The authors declare that no competing interests exist.

**Abstract** Alkaloids are important bioactive molecules throughout the natural world, and in many animals they serve as a source of chemical defense against predation. Dendrobatid poison frogs bioaccumulate alkaloids from their diet to make themselves toxic or unpalatable to predators. Despite the proposed roles of plasma proteins as mediators of alkaloid trafficking and bioavailability, the responsible proteins have not been identified. We use chemical approaches to show that a ~50 kDa plasma protein is the principal alkaloid-binding molecule in blood of poison frogs. Proteomic and biochemical studies establish this plasma protein to be a liver-derived alkaloid-binding globulin (ABG) that is a member of the serine-protease inhibitor (serpin) family. In addition to alkaloid-binding activity, ABG sequesters and regulates the bioavailability of 'free' plasma alkaloids in vitro. Unexpectedly, ABG is not related to saxiphilin, albumin, or other known vitamin carriers, but instead exhibits sequence and structural homology to mammalian hormone carriers and amphibian biliverdin-binding proteins. ABG represents a new small molecule binding functionality in serpin proteins, a novel mechanism of plasma alkaloid transport in poison frogs, and more broadly points toward serpins acting as tunable scaffolds for small molecule binding and transport across different organisms.

## Editor's evaluation

Poison frogs contain alkaloids in their plasma that make them toxic or unpalatable to predators, but how these animals avoid damage to themselves from their own defenses is not well understood. This valuable study identifies an alkaloid-binding protein, a member of the serpin superfamily, in the plasma of poison frogs that may explain how these animals are able to sequester a diverse array of alkaloids. With a convincing series of experiments, the authors advance our knowledge of the roles of serpins in animal ecophysiology.

## Introduction

Alkaloids are nitrogenous small molecules that play important ecological and physiological roles throughout nature, one of which is mediating predator–prey interactions. Species across many taxa, including plants, insects, marine invertebrates, and vertebrates, have co opted alkaloids as chemical defenses, methods for hunting, and pheromone signals. Some of the most potent alkaloid toxins, including batrachotoxin (BTX), saxitoxin (STX), and tetrodotoxin (TTX), act specifically by affecting voltage-gated ion channels, leading to disruption of nerve and muscle cells (*Wright, 2001*; *Wang et al., 2007*; *Efimova et al., 2020*). In the blue-ringed octopus, *Hapalochlaena lunulata*, TTX is used to paralyze prey (*Asakawa et al., 2019*), while in the pufferfish *Takifugu niphobles* it also acts as a pheromone (*Matsumura, 1995*), and in the California newt *Taricha torosa* it is a defense against predation (*Bucciarelli et al., 2014*). Other less-potent alkaloids also play important roles in predator–prey interactions. For example, Lepidoptera insects (butterflies and moths) and Coleoptera beetles sequester pyrrolizidine alkaloids from plants for predation, defense, and production of pheromones (*Hartmann et al., 1999*; *Cogni et al., 2012*). Although the identities of these alkaloids are well documented, less is known about the physiological mechanisms that allow animals to produce, sequester, and autoresist these small molecules. Identifying and characterizing proteins that interact with alkaloids in ecological contexts allow us to better understand how animal physiology has coevolved with alkaloids.

Despite the important ecological and physiological roles of alkaloids in animals, the molecular mechanisms involved in alkaloid production, transport, and resistance have been elusive and typically focused on a single alkaloid or specific structural class of alkaloids. In grasshoppers and moths, passive absorption of pyrrolizidine alkaloids is followed by conversion into a non-toxic form by hemolymph flavin-dependent monooxygenase, allowing the insects to avoid autointoxication (*Wang et al., 2012*). In some beetle species, ATP-binding cassette transporters actively pump pyrrolizidine alkaloids into reservoir defensive glands (*Strauss et al., 2013*). In vertebrates, the proteins that allow for the sequestration of alkaloids without autotoxicity are unclear with the exception of previous work with tetrodotoxin (TTX) and saxitoxin (STX). The pufferfish saxitoxin- and tetrodotoxin-binding protein (PSTBP) was originally identified in the plasma of *Fugu pardalis* (*Yotsu-Yamashita et al., 2001*), and is thought to play a role in the transport of TTX and STX to the site of bioaccumulation in the liver and ovaries in many pufferfish species (*Yotsu-Yamashita et al., 2018*). The soluble protein saxiphilin has been proposed as a toxin sponge for STX in various species of amphibians (*Abderemane-Ali et al., 2021*), although it remains unclear whether these species come into contact with STX in nature and whether saxiphilin acts as the predominant STX transporter in the plasma of frogs. While these insights have advanced our understanding of toxin physiology, studies in vertebrates have been narrowly focused on a few potent alkaloids with high-specificity binding proteins. The field is lacking a deeper molecular understanding of how certain species are able to accumulate multiple structurally diverse alkaloids for chemical defense.

Some species of frogs sequester a remarkable diversity of dietary alkaloids onto their skin as a chemical defense. This trait has independently evolved in several frog families, including Dendrobatidae in Central and South America and Mantellidae in Madagascar. Over 500 compounds have been found on the skin of Dendrobatidae frogs, with some alkaloids sourced from ants, mites, millipedes, and beetles (*Daly et al., 2002*; *Daly et al., 2005*; *Saporito et al., 2009*). Within dendrobatids, alkaloid-based chemical defenses have evolved independently at least three times (*Santos et al., 2003*; *Summers, 2003*), where non-toxic species do not uptake alkaloids onto their skin even when they are present in the diet (*Daly et al., 1994*; *Caldwell, 1996*; *Darst et al., 2005*; *Toft, 1980*). Well-studied poison frog alkaloids include pumiliotoxins (PTX), which targets sodium and potassium ion channels (*Vandendriessche et al., 2008*; *Daly et al., 1990*), and decahydroquinolines (DHQ), which affect nicotinic acetylcholine receptors (*Okada et al., 2021*). Epibatidine was first identified in the genus *Epipedobates* and specifically binds certain nicotinic receptors, leading it to be proposed as an analgesic alternative to morphine (*Spande et al., 1992*). Although there is limited research into the mechanisms of sequestration and autoresistance of alkaloids in poison frogs (*Abderemane-Ali et al., 2021*; *Caty et al., 2019*; *Alvarez-Buylla et al., 2022*; *O'Connell et al., 2021*; *Tarvin et al., 2017*), it is likely this process involves alkaloid transport through circulation for these dietary compounds to end up in skin storage glands. Based on the extensive work on plasma small molecule transport in mammals, one might expect that proteins like albumin, which is an abundant and promiscuous small molecule binder in the blood (*Peters, 1995*; *Baker, 2002*; *Czub et al., 2020*), or vitamin transporters

(*Haddad et al., 1993*; *Hall, 1975*; *Kanai et al., 1968*), which are able to interact with diet derived molecules, might be involved in alkaloid sequestration in poison frogs. In this study, we tested the hypothesis that poison frogs have an alkaloid-binding protein in the plasma and aimed to uncover its functional role and evolutionary significance. We predicted this protein would bind a range of poison frog alkaloids and would be present in frogs that are chemically defended in nature, but not in undefended species.

## Results

### An alkaloid-like photoprobe identifies a binding protein in poison frog plasma

We used a biochemical strategy to directly test for alkaloid binding in the plasma of different species of poison frogs. To do this, we obtained a UV crosslinking probe with an indolizidine functional group that shares structural similarity to the poison frog alkaloid pumiliotoxin **251D** (PTX) (*Figure 1A*, functional group highlighted in blue). Upon UV irradiation, the diazirine (green) enables protein crosslinking and the subsequent probe–protein complex can be conjugated to a fluorophore for gel-based visual analysis or biotin for streptavidin enrichment. Application of this photocrosslinking approach outside of mammalian systems has been remarkably limited, and in frogs has been limited to studying neuromuscular receptors (*Kiefer et al., 1970*; *Shinozawa et al., 1987*). We found the PTX-like photoprobe shows binding activity within the plasma of three species of dendrobatid poison frogs, *Oophaga sylvatica*, *Dendrobates tinctorius*, and *Epipedobates tricolor* (*Figure 1B*). In these species, this binding activity was restricted to a few bands in the 50–60 kDa range. Plasma from a non-toxic dendrobatid poison frog (*Allobates femoralis*), a mantellid poison frog (*Mantella aurantiaca*), the cane toad (*Rhinella marina*), and humans showed no binding activity with the photoprobe (*Figure 1B*). We further tested whether the presence of alkaloids would compete off photoprobe binding. In *O. sylvatica*, photoprobe binding resulted in two bands that showed competition by the addition of PTX, decahydroquinoline (DHQ), or epibatidine (epi), but not with nicotine (*Figure 1C*). In *D. tinctorius*, the photoprobe showed a two-band binding pattern, where the bottom band was competed by PTX and there was slight competition of both bands with DHQ and epibatidine, but no competition with nicotine (*Figure 1D*). In *E. tricolor* plasma two bands were observed, and these were more faint in the presence of PTX or DHQ, but not epibatidine or nicotine (*Figure 1E*). In both *O. sylvatica* and *D. tinctorius*, competition occurred when PTX was 10-fold higher in concentration than the photoprobe (*Figure 1—figure supplement 1*, *Figure 1—figure supplement 2*). We conclude from these photocrosslinking experiments that plasma binding of alkaloids in three species of chemically defended dendrobatid poison frogs is mediated by a ~50–60 kDa plasma protein.

### Proteomic analysis identifies an alkaloid-binding globulin

To identify the alkaloid-binding protein found in the plasma assays, we used *O. sylvatica* plasma to perform a pulldown and gel-punch proteomics on three conditions: no photoprobe (negative control), photoprobe only (positive control), and photoprobe with PTX competitor (*Figure 2A*). A biotin handle, instead of the fluorophore used above, was chemically added to the photoprobe for the enrichment of proteins using streptavidin beads. We used an untargeted proteomics approach to quantify and compare these enriched fractions using a proteome reference created from the *O. sylvatica* genome. Plasma from five *O. sylvatica* individuals was pooled to ensure sufficient quantity of protein for comparison. On average, 3876 unique peptides were found per sample, mapping to 433 *O. sylvatica* proteins (*Figure 2B*). The most highly abundant protein in the photoprobe condition had an average number of peptide spectral counts of 1224.5 and was competed off in the photoprobe with PTX condition by 64% (*Figure 2B*), resembling background levels (*Figure 2C*). This protein is annotated as serine-protease inhibitor A1 (serpinA1), which encodes for the protein alpha-1-antitrypsin (A1AT). As our subsequent experiments demonstrate this protein functions as an alkaloid binding and sequestration protein, we refer to it hereafter as 'alkaloid-binding globulin' (ABG). Mapping the 72 unique peptides onto the protein sequence of ABG showed full coverage across the protein, excluding the signal peptide, which in other serpins is cleaved during secretion (*Figure 2—figure supplement 1*). In comparison to ABG, albumin showed high abundance but no competition (*Figure 2B, D*). We conclude that ABG functions as a major alkaloid-binding protein in poison frog plasma.

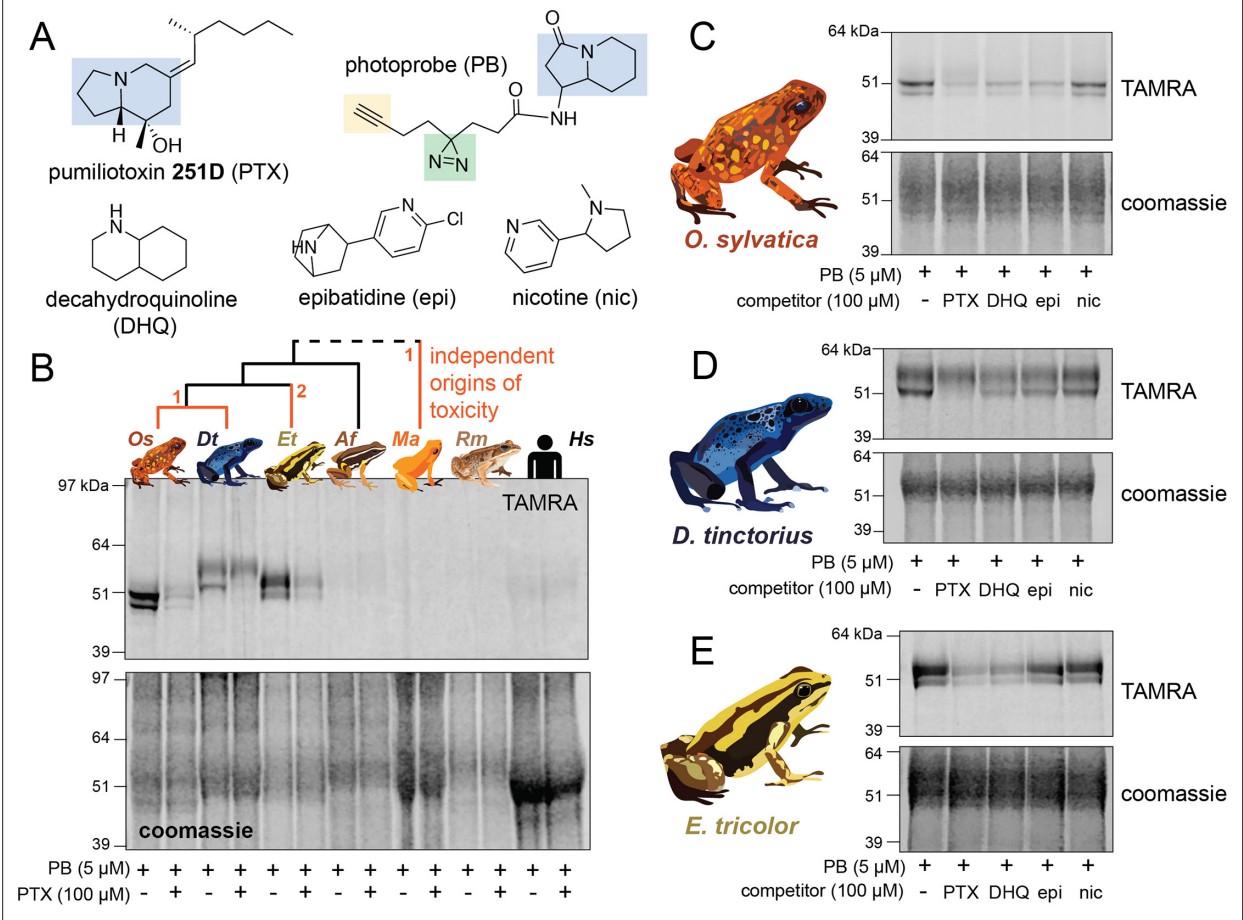

**Figure 1.** Alkaloid-like photocrosslinking probes show binding and competition in poison frog plasma. (**A**) Structures of alkaloid-like photocrosslinking probe and alkaloids tested, with the functional group in blue, the diazirine group in green, and the terminal alkyne in yellow. In (**B–E**), the top images show the TAMRA signal, which visualizes photoprobe binding, and the bottom images show coomassie staining of the same gel to assess total protein concentration in each well. (**B**) Plasma from different species (*Oophaga sylvatica – Os*, *Dendrobates tinctorius – Dt*, *Epipedobates tricolor – Et*, *Allobates femoralis – Af*, *Rhinella marina – Rm*, and humans – *Hs*, from left to right) shows different plasma photoprobe-binding activity and competition. Orange lines on phylogeny indicate independent evolutionary origins of chemical defense in Dendrobatidae and Mantellidae, with the number representing the number of times the phenotype arose along that branch. (**C**) *Oophaga sylvatica* plasma shows crosslinking, and competition with pumiliotoxin (PTX), decahydroquinoline (DHQ), and epibatidine (epi), but not nicotine (nic). (**D**) *Dendrobates tinctorius* plasma shows crosslinking and competition with PTX, slight competition with DHQ and epi, and no competition with nic. (**E**) *Epipedobates tricolor* plasma shows crosslinking and competition with PTX and DHQ, but not with epi or nic.

The online version of this article includes the following source data and figure supplement(s) for figure 1:

Source data 1. Raw data for the gels shown in *Figure 1*.

Figure supplement 1. *O. sylvatica* dose response of photoprobe competition.

Figure supplement 1—source data 1. Raw data for the gel shown in *Figure 1—figure supplement 1*.

Figure supplement 2. *D. tinctorius* dose response of photoprobe competition.

Figure supplement 2—source data 1. Raw data for the gel shown in *Figure 1—figure supplement 2*.

## Structural predictions of ABG show binding pocket similarities to mammalian hormone carriers

The identification of ABG as the principal alkaloid-binding protein in plasma was unexpected, as plasma binding of small molecules is commonly mediated by albumin. Nevertheless, in mammals, members of the serpinA family function as carriers of lipophilic hormones, providing plausibility to the hypothesis that frog serpin protein family members may also bind small molecules. Therefore, we sought further structural insights into ABG using protein structure predictions and molecular docking simulations to examine if this protein has a predicted binding pocket for small molecules. Using AlphaFold to predict

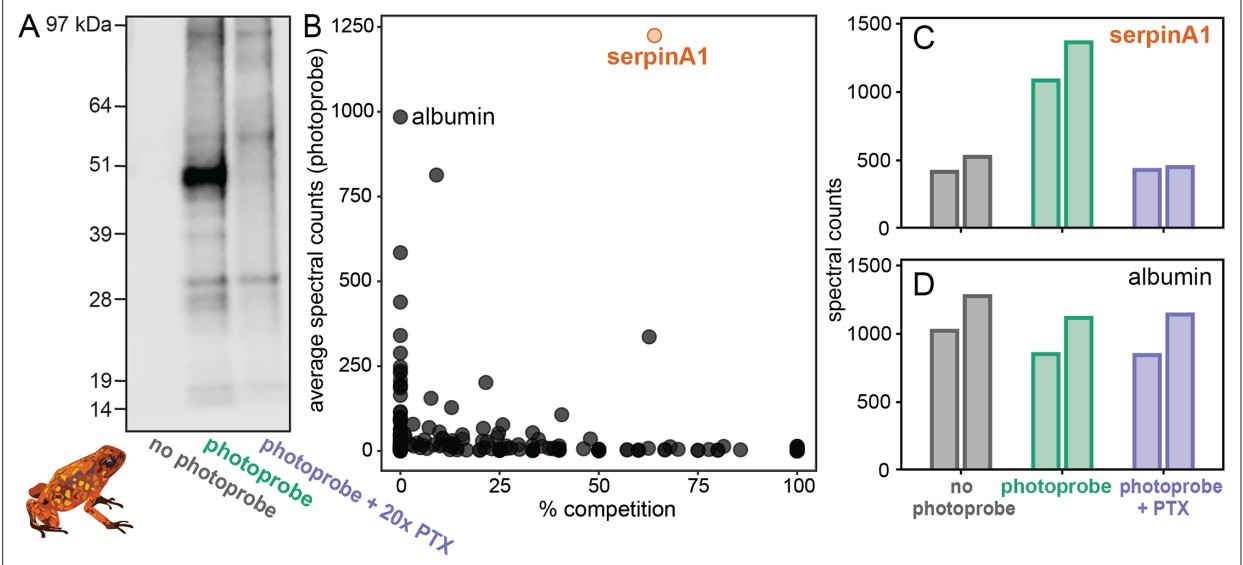

**Figure 2.** Proteomics identifies a serpinA1-like protein as the main pumiliotoxin (PTX)-binding protein in *Oophaga sylvatica* plasma. (**A**) Streptavidin blot of the proteins pulled down from *O. sylvatica* plasma across the three conditions: no photoprobe, photoprobe, and photoprobe plus competitor PTX. (**B**) Quantitative proteomics output in terms of percent competition defined as 100% − average spectral counts in the photoprobe + PTX condition divided by average spectral counts in the photoprobe only condition. Average was taken across two replicates. SerpinA1-like protein and albumin are annotated. (**C**) The number of spectral counts across conditions for the serpinA1-like protein, each replicate is shown individually. (**D**) The number of spectral counts across conditions for the albumin protein, each replicate is shown individually.

The online version of this article includes the following source data and figure supplement(s) for figure 2:

**Source data 1.** Raw data for the blot and proteomics shown in *Figure 2*.

**Figure supplement 1.** Peptide coverage over alkaloid-binding globulin (ABG) protein sequence.

**Figure supplement 1—source data 1.** Raw data for the proteomics shown in *Figure 2—figure supplement 1*.

the structure of the full protein sequence without the signal peptide resulted in a high confidence structure (*Figure 3A*). We then compared it to the structures of serpinA6/corticosteroid-binding globulin (CBG, *Figure 3B*; *Klieber et al., 2007*), biliverdin-binding serpin (BBS, *Figure 3C*; *Manoilov et al., 2022*), serpinA1/alpha-1-antitrypsin (A1AT, *Figure 3—figure supplement 1*; *Kim et al., 2001*), and serpinA7/thyroxine-binding globulin (TBG, *Figure 3—figure supplement 2*; *Qi et al., 2011*). The AlphaFold prediction for *O. sylvatica* ABG (*Os*ABG) demonstrated a conserved structural element of three alpha helices backed by a set of beta sheets, which is the small molecule binding pocket in CBG, BBS, and TBG (black boxes, *Figure 3A–C*, *Figure 3—figure supplement 1*, *Figure 3—figure supplement 2*), and also exists in the non-small molecule binding A1AT (black box, *Figure 3—figure supplement 1*). When a molecular docking simulation was run with the whole *Os*ABG protein as the search space and PTX as the ligand, the highest affinity binding site was in the same binding pocket defined by this structural motif (*Figure 3D*). Although the overall structural components of the binding pockets show similarities across *Os*ABG, CBG, BBS, and TBG, the individual amino acids coordinating small molecule binding differ across proteins (*Figure 3D–F*, *Figure 3—figure supplement 2*, *Figure 3—figure supplement 3*). These results offer a structural explanation for PTX binding by ABG and highlight the homology between ABG and other small molecule binding serpins.

## Recombinant expression recapitulates binding activity of different ABG proteins

To confirm ABG-binding activity in vitro and compare across different species, we recombinantly expressed and purified *O. sylvatica* ABG (*Os*ABG), and its closest homolog from the *D. tinctorius* and *E. tricolor* transcriptomes (*Dt*ABG and *Et*ABG, respectively). The resulting purified protein doublet is due to post-translational glycosylation differences, as *Os*ABG has two predicted *N*-glycosylation sites (*Gupta and Brunak, 2002*) and the doublet disappears when treated with a glycosylase enzyme (*Figure 4—figure supplement 1*). As expected, purified *Os*ABG recapitulated the binding

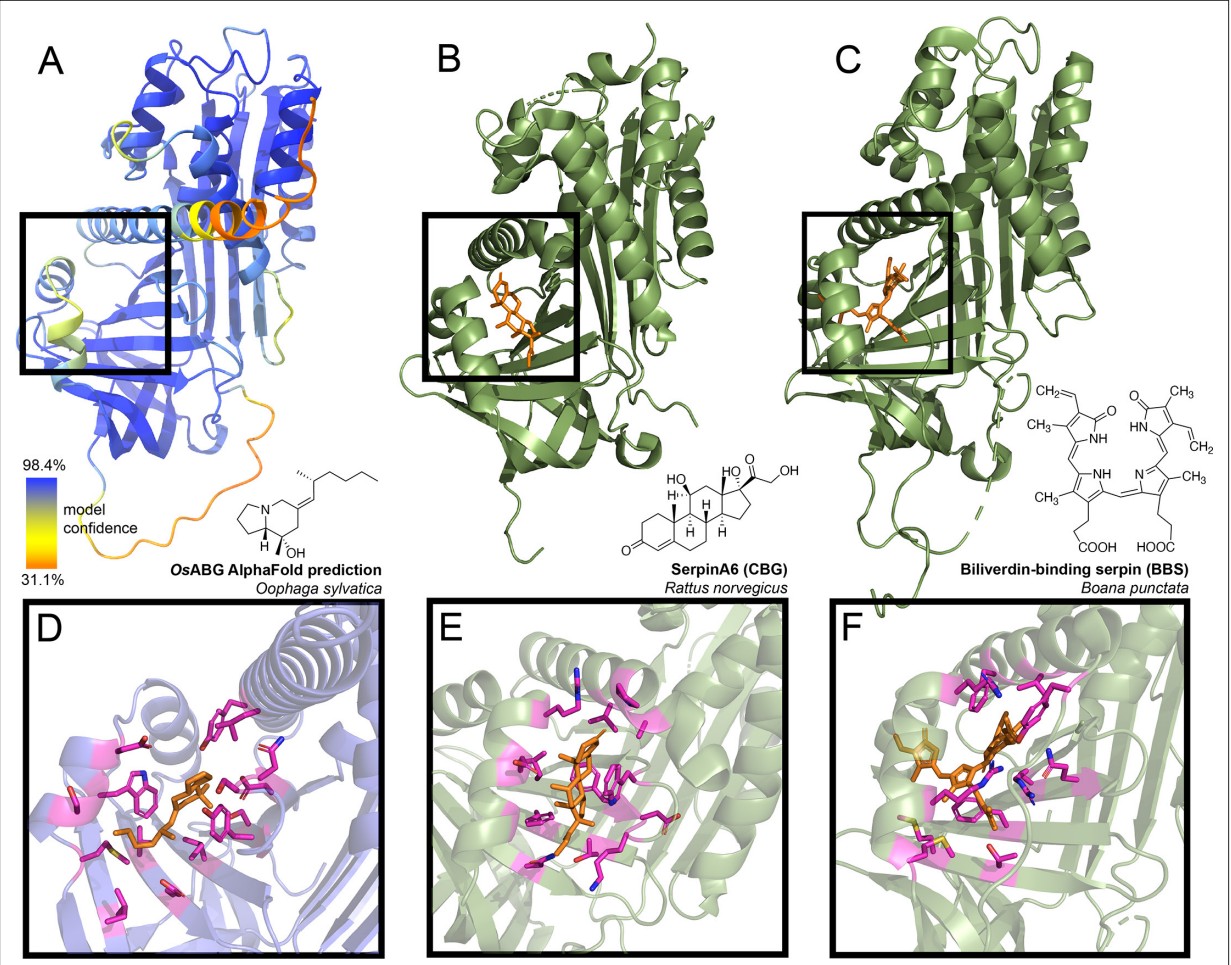

**Figure 3.** Predicted alkaloid-binding globulin (ABG) structure and binding pocket resembles that of other small molecule binding serpins. (**A**) AlphaFold structure predicted with the protein sequence of the *Oophaga sylvatica* ABG, with color representing model confidence and predicted binding pocket based on molecular docking simulation indicated with a black box. (**B**) Crystal structure for rat SerpinA6/corticosteroid-binding globulin (CBG), with the cortisol molecule shown in orange (PDB# 2V95). (**C**) Crystal structure for tree frog *Boana punctata* biliverdin-binding serpin (BBS), with biliverdin shown in orange (PDB# 7RBW). (**D**) Close-up of predicted binding pocket of pumiliotoxin (PTX) in *O. sylvatica* ABG, with residues proximal to PTX highlighted in magenta. The structure of PTX is indicated on the top right. (**E**) Close-up of cortisol binding in CBG (PDB# 2V95), with proximal residues highlighted in magenta. Cortisol structure is displayed on the top right. (**F**) Close-up of biliverdin binding in BBS (PDB# 7RBW), with some proximal residues highlighted in magenta. Biliverdin structure is shown on the top right.

The online version of this article includes the following source data and figure supplement(s) for figure 3:

**Source data 1.** Raw data for the structure prediction shown in *Figure 3*.

**Figure supplement 1.** Structure of A1AT protein.

**Figure supplement 2.** Structure and binding pocket of thyroxine-binding globulin (TBG) protein.

**Figure supplement 3.** Alignment of binding pocket residues.

**Figure supplement 3—source data 1.** Raw data for the alignment shown in *Figure 3—figure supplement 3*.

and competition seen with the plasma, where the photoprobe was most fully competed by the presence of PTX, and also competed off by DHQ and epibatidine, but not nicotine (*Figure 4A*). The competition activity with the purified protein was noticeable at a ratio of one to one photoprobe to PTX (*Figure 4—figure supplement 2*). Purified *Dt*ABG and *Et*ABG required higher concentrations of protein to see a signal and showed much weaker photoprobe binding, which was competed off by the presence of PTX and DHQ for both *Dt*ABG (*Figure 4B*) and *Et*ABG (*Figure 4C*). Together these results confirm the plasma findings that ABG is a multi-alkaloid-binding protein with different specificities and

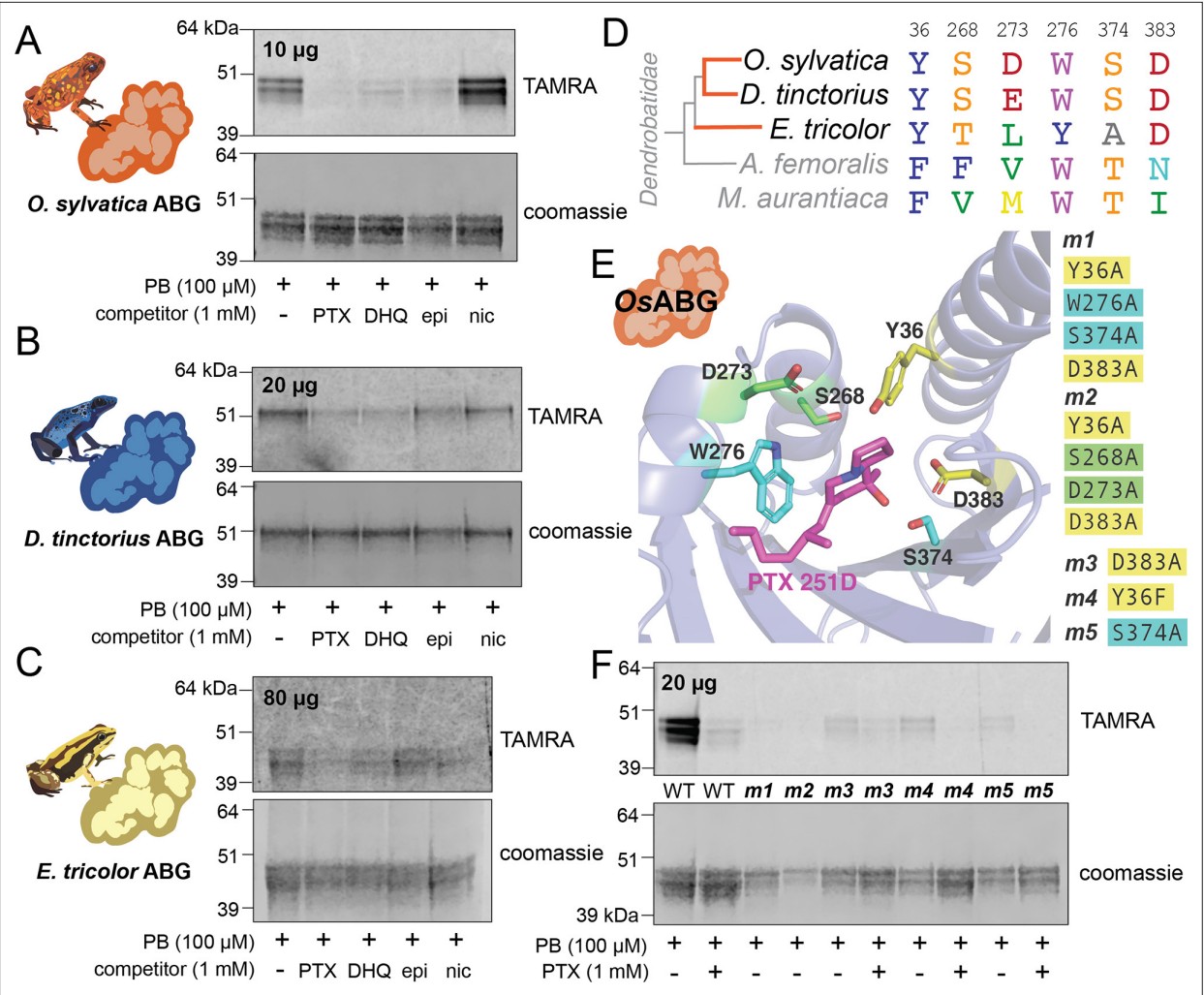

**Figure 4.** Recombinant expression and binding pocket mutants confirm plasma-binding activity and binding pocket predictions. (**A**) Photoprobe crosslinking and competition with different compounds of 10 µg recombinantly expressed and purified *Os*ABG recapitulates the binding activity seen in the plasma (*Figure 1C*). (**B**) Photoprobe crosslinking with 20 µg recombinantly expressed *Dendrobates tinctorius* alkaloid-binding globulin (ABG) shows crosslinking, and competition with pumiliotoxin (PTX) and decahydroquinoline (DHQ). (**C**) Photoprobe crosslinking with 80 µg recombinantly expressed *Epipedobates tricolor* ABG shows crosslinking, and competition with PTX and DHQ. (**D**) Alignment of protein sequence of proteins homologous to *Os*ABG across species shows conservation of certain amino acids. Coloring of amino acids is based on the RasMol 'amino' coloring scheme, which highlights amino acid properties. (**E**) Potential binding residues were identified from the molecular docking simulation. Five different mutants were made based on specific amino acids in the binding pocket, with either a combination of four different alanine substitutions (m1 – yellow and teal residues, and m2 – yellow and green residues) or a single substitutions at D383 (m3), Y36F (m4), or S374A (m5). P8TX is shown in magenta. Oxygen atoms on the molecules are highlighted in red, nitrogen in blue. (**F**) Quadruple binding pocket mutants (m1 and m2) lose binding activity of the photoprobe, single amino acid substitutions (m3, m4, and m5) show reduced photoprobe binding and retained competition with PTX.

The online version of this article includes the following source data and figure supplement(s) for figure 4:

**Source data 1.** Raw data for the gels shown in *Figure 4*.

**Figure supplement 1.** Glycosylation of recombinant *Os*ABG.

**Figure supplement 1—source data 1.** Raw data for the coomassie gel shown in *Figure 4—figure supplement 1*.

**Figure supplement 2.** Dose response of crosslinking *Os*ABG.

**Figure supplement 2—source data 1.** Raw data for the gels shown in *Figure 4—figure supplement 2*.

**Figure supplement 3.** Recombinant expression and purification of alkaloid-binding globulin (ABG) proteins in insect cells.

**Figure supplement 3—source data 1.** Raw data for the blots and gels shown in *Figure 4—figure supplement 3*.

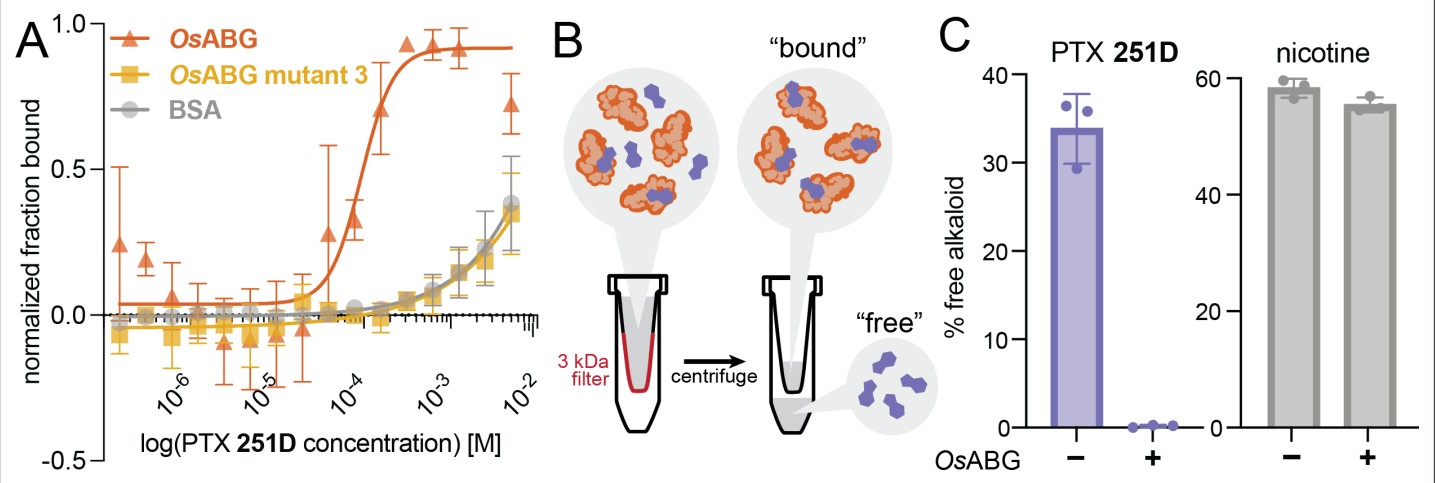

**Figure 5.** *Os*ABG sequesters free pumiliotoxin (PTX) in solution. (**A**) Microscale thermophoresis (MST) of labeled *Os*ABG with PTX finds that binding is of higher affinity than that of *Os*ABG mutant 3 (D383A) and bovine serum albumin (BSA). (**B**) A 3-kDa molecular weight cut off (MWCO) centrifuge filter was used to separate 'free' versus 'bound' alkaloids in solutions with and without *Os*ABG present, to later be quantified with liquid chromatography–mass spectrometry (LC–MS). (**C**) The percent of 'free' PTX 251D (purple) dropped when *Os*ABG was present, however the amount of 'free' nicotine (gray) remained unchanged by the presence of *Os*ABG.

The online version of this article includes the following source data for figure 5:

**Source data 1.** Raw data for the microscale thermophoresis (MST) and liquid chromatography–mass spectrometry (LC–MS) data shown in *Figure 5*.

---

affinities across poison frog species, and that *Os*ABG alone is sufficient to recapitulate the crosslinking activity observed in the plasma.

Given the predicted binding pocket from the molecular docking simulations, and the differences in binding activity of the ABG proteins in different poison frog species, we used a sequence (*Figure 4D*) and predicted structure (*Figure 4E*) informed approach to mutate residues that might coordinate alkaloid binding in the hypothesized pocket. We identified six residues with proximity to the docked PTX molecule that might have important binding activity: Y36, S268, D273, W276, S374, and D383 (*Figure 4E*). Mutating different sets of these binding residues in the *Os*ABG sequence led to a disruption of binding and competition. The combined mutations of Y36A, W276A, S374A, D383A (m1) and Y36A, S268A, D273A, D383A (m2) disrupted binding to the photoprobe completely (*Figure 4F*). The single-point mutations of D383A (m3), Y36F (m4), and S374A (m5) weakened photoprobe binding significantly, to the point of being nearly undetectable (*Figure 4F*) in comparison to the wild-type protein. All single-point mutations retained the ability to compete photoprobe binding with PTX (*Figure 4F*). These results demonstrate that mutating residues in the binding pocket identified through molecular docking disrupts binding activity of *Os*ABG, providing biochemical evidence that the structurally predicted binding pocket of ABG indeed is the relevant binding site for PTX. Furthermore, we have identified a set of residues that are necessary for PTX binding with high affinity, showing that the plasma-binding activity is coordinated by specific amino acids in *Os*ABG.

### *Os*ABG sequesters free PTX in solution with high affinity

Previous work has described small molecule binding serpins and their important role in regulating the pool of free versus bound ligands in circulation (*Chan et al., 2013*; *Pemberton et al., 1988*; *Gardill et al., 2012*; *Lewis et al., 2005*). We hypothesized that *Os*ABG might play a similar role for alkaloids in the poison frog plasma. To test this, we examined both the direct binding of *Os*ABG for PTX and its ability to regulate the pool of bioavailable alkaloids in solution. Using microscale thermophoresis (MST) we found that wild-type *Os*ABG binds PTX with greater affinity than bovine serum albumin (BSA) or *Os*ABG mutant 3 (*Figure 5A*). *Os*ABG mutant 3, D383A, had similar binding affinity for PTX as BSA (*Figure 5A*). To test the ability of *Os*ABG to sequester alkaloids in vitro, we used a 3-kDa molecular weight cutoff centrifuge filter to separate the 'bound' and 'free' PTX (*Figure 5B*), which we then quantified by liquid chromatography–mass spectrometry (LC–MS) We found that in the presence

of *Os*ABG, the amount of 'free' PTX is dramatically reduced, while that of nicotine is not (*Figure 5C*). These direct binding assay results show that *Os*ABG is able to bind PTX with high affinity, and therefore may regulate the amount of free PTX in solution. Regulation of bioavailable pools of PTX in circulation may have downstream consequences on sequestration, transcription, and signaling throughout the organism.

## *Os*ABG is highly expressed in wild frogs and binds ecologically relevant toxins

We next sought to better understand the functional role of *Os*ABG in a context relevant to the ecology and physiology of poison frogs. *O. sylvatica* frogs were collected across three different locations in Ecuador (*Figure 6A*). Tissue RNA sequencing revealed that *Os*ABG mRNA is expressed very highly in the liver compared to other tissues (*Figure 6—figure supplement 1*). Hierarchical clustering of all unique serpinA genes in the genome shows that *Os*ABG is most closely related to two other serpinA1 genes, *Os*4677 and *Os*4682 (*Figure 6B*). The liver expression of *Os*ABG is higher than all other serpinA genes, and is higher than the expression of albumin in the liver (*Figure 6B*). Field-collected *O. sylvatica* frog skin contains alkaloids from many different classes, with 33% of the summed alkaloid load being histrionicotoxins, followed by 22% in 5,8-indolizidines, 15% in 3,5-indolizidines, 13% in 5,6,8-indolizidines, and 10% in DHQ (*Figure 6C*). Further crosslinking experiments with purified *Os*ABG found that it also binds a histrionicotoxin-like base ring structure (HTX), indolizidine (indol), and shows slight competition by a toxin mixture created from wild frog skin extracts (*Figure 6D*). To understand the tissue distribution of *Os*ABG, we created a custom anti-*Os*ABG antibody and stained *O. sylvatica* intestines, skin, and liver (*Figure 6E, F*, *Figure 6—figure supplement 2*). We found *Os*ABG staining signal along the deeper layers of the intestinal mucosa and along the inner dermal layers underlying the skin granular glands (*Figure 6E*). When the anti-*Os*ABG antibody was pre-incubated with purified *Os*ABG protein prior to staining, the signal was lost (*Figure 6F*). Together, these data characterize the expression profile and distribution of *Os*ABG and show that it is capable of binding other alkaloid classes that are found in wild frogs.

## Discussion

ABG represents a new small molecule binding functionality for a member of the serpin family, with a structurally conserved binding pocket similar to mammalian hormone carriers and BBS. This provides evidence for convergent evolution of serpin proteins for the binding and transport of small molecules across taxa and physiological roles. Most serpin proteins are known for their protease inhibitory activity, however there are members of the serpin superfamily that have been instead characterized as non-inhibitory small molecule binding globulins. In tree frogs, BBS has been identified as the primary protein that binds biliverdin (*Taboada et al., 2020*), the heme metabolite that is responsible for green coloration in the lineage, although it remains unclear whether BBS is also responsible for the transport of biliverdin across tissues. In mammals, cortisol-binding globulin (CBG, serpinA6) and TBG (serpinA7) play important roles in the transport and regulation of plasma hormone concentrations (*Robbins, 1992*; *Siiteri et al., 1982*). However, serpins are not the only carriers of lipophilic small molecules in mammalian plasma, with many other hormones and vitamins moving through plasma on albumin (*Peters, 1995*; *Baker, 2002*), vitamin-D-binding protein (*Bouillon et al., 2019*; *Chun, 2012*), and sex-hormone-binding globulin (*Fortunati, 1999*; *Round et al., 2020*), which have no homology to serpin proteins. That albumin is not the primary carrier for PTX in poison frog plasma is surprising given its abundance and previously documented roles in small molecule transport, although it is possible that there are poison frog alkaloids that are transported on albumin instead of ABG. CBG and TBG have evolved small molecule transport from a protease inhibitor background (*Spence et al., 2021*), and the discovery of ABG shows that alkaloid binding in a serpin evolved either independently or from hormone transport in poison frogs. BBS, CBG, and TBG all bind their respective ligands in a similar structural pocket (*Manoilov et al., 2022*), however the identity of the residues that coordinate binding vary across the proteins. Mutations of key binding site residues in *Os*ABG were able to disrupt binding, confirming the functional homology to BBS, CBG, and TBG. A more detailed mutational scanning would be necessary to fully understand which residues coordinate the binding of different alkaloids, however we see that single-point mutations significantly disrupt binding ability (*Figures 4F*

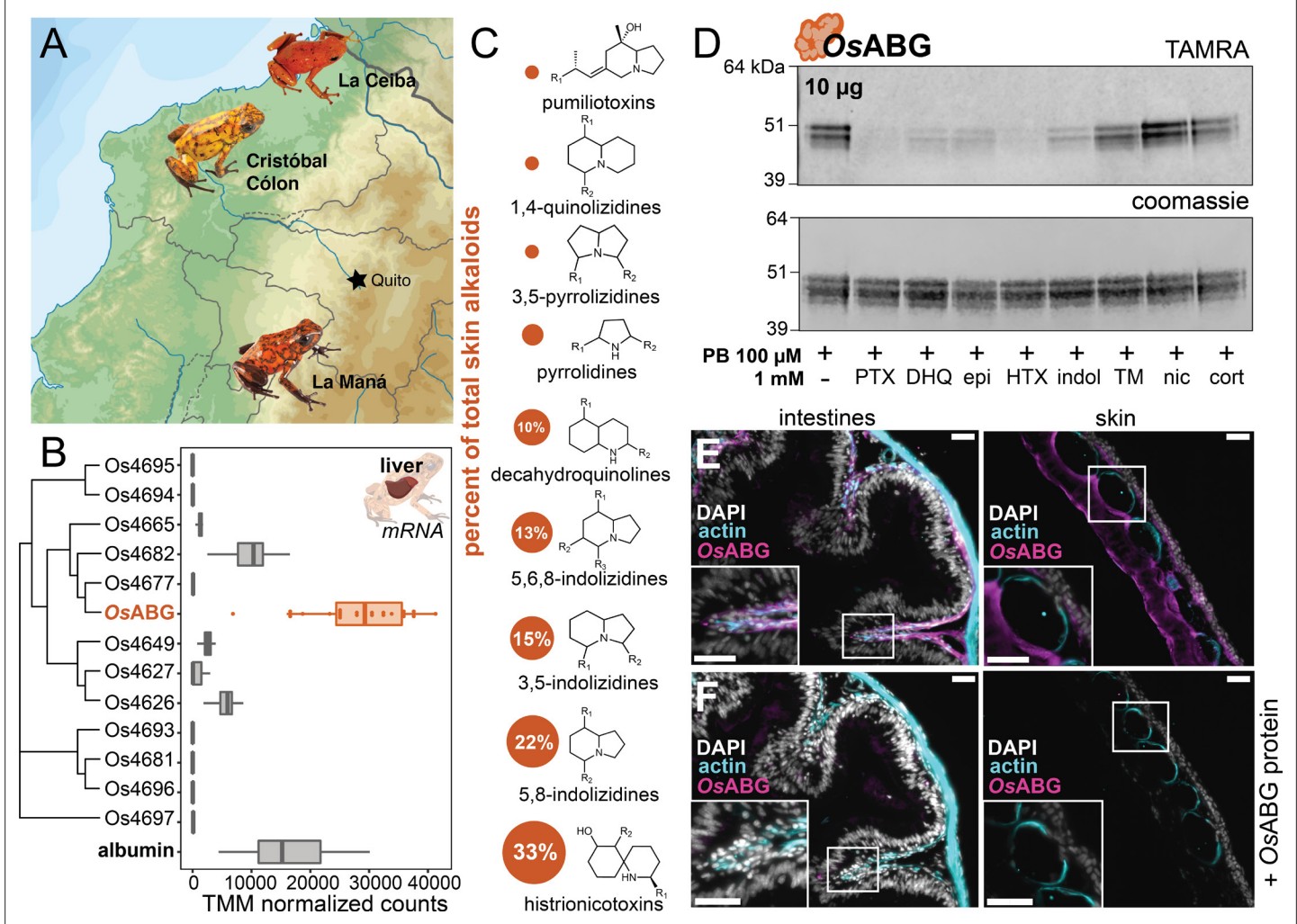

**Figure 6.** *Os*ABG is expressed in the liver and binds ecologically relevant alkaloids. (**A**) Wild *Oophaga sylvatica* were collected across three locations in Ecuador, *n* = 10 per location. (**B**) The liver expression level of *Os*ABG was higher than that of other members of the serpinA family found in the genome, and of albumin. (**C**) Dorsal skin alkaloids fell into nine different classes, with the size of the circle representing the averaged percent of total skin alkaloid load. (**D**) Photoprobe binding with recombinantly expressed *Os*ABG was competed by pumiliotoxin (PTX), decahydroquinoline (DHQ), epi, a histrionicotoxin-like compound (HTX), and indolizidine ring without R groups (indol), and slightly by a mixture of skin toxins from the wild specimens (TM). Photoprobe binding was not competed by nicotine (nic) or cortisol (cort). (**E**) Custom anti-*Os*ABG antibody staining (magenta) in the skin and intestines, with actin stain (blue) and 4′,6-diamidino-2-phenylindole (DAPI, shown in white). (**F**) Pre-incubation of anti-*Os*ABG with purified *Os*ABG protein in the skin and intestines shows loss of *Os*ABG staining, indicating specific staining activity. White bars represent 50 μm.

The online version of this article includes the following source data and figure supplement(s) for figure 6:

**Source data 1.** Raw data for the gene expression, gels, and immunohistochemistry shown in *Figure 6*.

**Figure supplement 1.** *Os*ABG expression in intestines, liver, and skin.

**Figure supplement 1—source data 1.** Raw data for the gene expression data shown in *Figure 6—figure supplement 1*.

**Figure supplement 2.** *Os*ABG protein presence in intestines, liver, and skin.

**Figure supplement 2—source data 1.** Raw data for the immunohistochemistry shown in *Figure 6—figure supplement 2*.

**Figure supplement 3.** Western blot of anti-*Os*ABG protein.

**Figure supplement 3—source data 1.** Raw data for the western blot shown in *Figure 6—figure supplement 3*.

*and 5A*), suggesting a carefully encoded specificity. It would be interesting to expand binding experiments to more classes of alkaloids, although we are limited by the commercial availability of many of these compounds. The similarities between ABG and mammalian hormone carriers raises the possibility that alkaloids may also act as signaling molecules, which would require further characterization

of other alkaloid targets in poison frogs and more mechanistic information about ABG's alkaloid transport. The identification of this novel alkaloid carrier opens up new avenues of research in the evolution of small molecule transporters and their relevant physiological functions, especially in amphibians. The data presented suggest that *Os*ABG is using a binding site already present in serpin family proteins, and has encoded remarkable specificity for certain alkaloids within this site.

ABG has different binding specificity across poison frog species and independent evolutionary origins of acquired chemical defense, which supports the idea that there are conserved mechanisms of alkaloid sequestration across the poison frog lineage. Of the species tested, the photoprobe showed binding activity only in dendrobatid species that can acquire alkaloid chemical defenses from their diet, namely *O. sylvatica*, *D. tinctorius*, and *E. tricolor*, which represent two independent origins of chemical defense (*Santos et al., 2003*). This suggests that plasma proteins have evolved in dendrobatid frogs that are capable of acquired chemical defense, as binding was not seen in the dendrobatid without chemical defenses in nature (*A. femoralis*), the Malagasy poison frog (*M. aurantiaca*), cane toads, or human plasma. Malagasy poison frogs have convergently evolved many of the notable phenotypes found in the dendrobatid lineage, and have been found to contain PTX and other alkaloids (*Garraffo et al., 1993*; *Fischer et al., 2019*; *Clark et al., 2005*). The lack of binding activity in the plasma may indicate that Malagasy poison frogs have different molecular mechanisms for alkaloid transport and autoresistance. Future characterization of plasma-binding activity with more species and alkaloid compounds would be critical to fully trace the evolution of ABG binding across the poison frog lineage.

Using competition of the photoprobe by presence of excess alkaloid as a proxy for binding activity, we found differences in the competition activity of different alkaloids across dendrobatid species. This different plasma binding suggests that *O. sylvatica* has a more promiscuous alkaloid-binding pocket, which may be related to the high diversity of alkaloids found in wild *O. sylvatica* frogs (*Caty et al., 2019*; *McGugan et al., 2016*; *Santos et al., 2016*). In *D. tinctorius* and *E. tricolor* plasma the photo-crosslinking banding pattern and size of the bands was different than that of *O. sylvatica*. We hypothesize that these plasma bands consist of the ABG proteins identified through homology (*Figure 4B ,C*), however because we did not perform proteomics quantification, the possibility remains of other alkaloid-binding proteins in *D. tinctorius* and *E. tricolor*, which would suggest that ABG is an evolutionary innovation unique to the *O. sylvatica* lineage. The identification of *Os*ABG opens up the possibility of further discoveries of ABGs in other toxic species. Overall, the diversity in plasma binding seen across phylogenetically close species may reflect the diversity in environmental pressures that have led to species-specific adaptations at the molecular scale.

*Os*ABG is expressed at high levels and binds additional alkaloids found in high abundance on the skin of field-collected *O. sylvatica*, indicating its physiological and ecological relevance for this species. *Os*ABG is produced in the liver as is the case with most members of the serpin family in humans (*Fagerberg et al., 2014*), cows (*Merkin et al., 2012*), rats (*Yu et al., 2014*), baboons (*Pipes et al., 2013*), and macaques (*Merkin et al., 2012*). The liver expression level of *Os*ABG was higher than other serpinA proteins and albumin, which in most vertebrates is the most abundant plasma protein (*Baker, 2002*; *Doweiko and Nompleggi, 1991*; *Merlot et al., 2014*), further highlighting the importance of this protein for poison frog physiology. Additionally, we were able to detect the presence of *Os*ABG protein with a custom antibody in the inner mucosal layers of the plicae circulares of the intestines (*Smith et al., 2000*) and in the the compact dermis of the skin (*Varga et al., 2018*), which surrounds the granular glands where poison frogs store their alkaloid reserves (*Neuwirth et al., 1979*). Taken together, this supports a model where *Os*ABG is produced in the liver, binds alkaloids present in the blood or recently absorbed by the intestines, and then transports alkaloids to the skin for bioaccumulation. Previous work with CBG and TBG in mammals shows that they bind 70–90% of their respective ligands in the plasma, thus regulating the concentration of free hormone in the blood (*Siiteri et al., 1982*; *Refetoff, 2003*). Furthermore, CBG and TBG have transport activity, as the release of the ligand is induced by proteolytic cleavage (*Robbins, 1992*; *Hammond et al., 1990*). ABG may be acting in a similar way to mediate the bioaccumulation of dietary alkaloids onto the skin. The reduction of 'free' PTX when *Os*ABG is present in vitro further supports this, given that *Os*ABG would be regulating the pool of unbound alkaloids available in the plasma. Further identification and characterization of the respective protease from *O. sylvatica* would be necessary to validate this hypothetical mechanism for alkaloid release and genetic knockout frogs are needed to fully characterize

the organismal role of *Os*ABG. Given our findings we conclude that ABG may play an important transport role in poison frog physiology and chemical defense, which greatly broadens our understanding of this previously elusive mechanism in poison frogs. Furthermore, given its striking similarities with mammalian hormone carriers and BBS, our findings open up additional questions regarding the functional convergence of serpins as small molecule transporters in distantly related taxa.

## Summary

This study presents the first evidence of an alkaloid-binding protein in poison frog plasma with a suggested role as a transporter molecule. We found a novel alkaloid-binding function for a member of the serpin family, contributing to a mounting body of evidence suggesting that the small molecule binding activity of certain serpins has evolved multiple times from their protease inhibitor precursors with a structurally conserved binding pocket. Our alkaloid comparisons with *Os*ABG show that its specificity has been fine-tuned for certain small molecule substrates, and it may play an important ecological and physiological role in the evolution of chemical defense in poison frogs. The discovery of a serpin that can bind a number of different ligands sets up the possibility of future efforts to understand and engineer ABG's binding pocket for new ligands once an in-depth understanding of the binding specificity is achieved.

# Materials and methods

## Animal usage

All animal procedures were approved by the Institutional Animal Care and Use Committee at Stanford (protocol #34153). Topical benzocaine was used for anesthesia prior to the euthanasia of all animals. Laboratory-bred animals were purchased from Understory Enterprises (Ontario, Canada) or Josh's Frogs (Michigan, USA) depending on the species. Animals were either euthanized for plasma collection upon arrival, or housed in $18^3$ inch glass terraria, and fed a diet of non-toxic *Drosophila melanogaster* until euthanasia. Plasma and tissues from a total of 62 animals were used for this study, consisting of 32 lab-bred animals and 30 field-collected animals that are described below. Sample size of field-collected animals was determined based on variability seen in previous studies, and for laboratory experiments based on experimental needs in terms of volume of plasma.

## Plasma collections

Lab-bred, and therefore non-toxic, poison frogs were anesthetized with topical application of 10% benzocaine on the ventral skin, and euthanized with cervical translocation. Blood was collected directly from the cervical cut using a heparinized capillary tube (22-362-566, Fisher Scientific, Massachusetts, USA) and deposited into a lithium heparin coated microvette tube (20.1282.100, Sarstedt, Nümbrecht, Germany). Blood was spun down 10 min at 5000 rpm at 4°C on a benchtop centrifuge, and the top layer which contains the plasma was removed and pipetted into a microcentrifuge tube. This was stored at −80°C until it was used for experiments.

## UV crosslinking and competition using alkaloid-like photoprobe

Photocrosslinking experiments followed methods outlined in *Kim et al., 2020*. Plasma or purified protein was thawed on ice. The total reaction volume was 50 µl and all experiments were performed in a clear 96-well plate. For plasma, 5 µl of undiluted plasma was mixed with 40 µl of phosphate-buffered saline (PBS) for each reaction. For purified protein, varying amounts of protein (amount indicated on each gel image) were diluted into PBS per reaction to a volume of 45 µl. For *E. tricolor* and *D. tinctorius* protein a higher amount of protein was used because no photocrosslinking was detected at 10 µg for these proteins. To this, either 2.5 µl of dimethyl sulfoxide (DMSO) was added as a vehicle control, or 2.5 µl of competitor compound dissolved in DMSO was added at the concentrations indicated in the figure panels. The competitor compounds were: custom synthesized PTX **251D** (PepTech, Massachusetts, USA), DHQ (125741, Sigma-Aldrich, Missouri, USA), epibatidine (epi, E1145, Sigma-Aldrich), a histrionicotoxin-like compound (HTX, ENAH2C55884A-50MG, Sigma-Aldrich), indolizidine (indol, ATE24584802-100MG, Sigma-Aldrich), nicotine (nic, N3876-100ML, Sigma-Aldrich), and cortisol (cort, H0888-1G, Sigma-Aldrich). The 'toxin mixture' (TM) used as a competitor in *Figure 6* was made by taking 20 µl of each of the skin alkaloid extracts from wild frogs described below, evaporating it

under gentle nitrogen gas flow, and resuspending in 100 µl of DMSO. After adding the competitor compound, 2.5 µl of photoprobe (Z2866906198, Enamine, Kyiv, Ukraine) dissolved in DMSO was added to the reaction on ice, for a final photoprobe concentration of 5 µM in plasma experiments and 100 µM in purified protein experiments. This was incubated on ice for 10 min, and then UV crosslinked (Stratalinker UV 1800 Crosslinker, Stratagene, California, USA) for 5 min on ice. TAMRA visualization of crosslinked proteins was done by adding 3 µl tris(benzyltriazolylmethyl)amine (TBTA) (stock solution: 1.7 mM in 4:1 vol/vol DMSO:tert-Butanol; H66485-03, Fisher), 1 µl copper (II) sulfate (stock solution: 50 mM in water; BP346-500, Fisher), 1 µl tris (2-carboxyethyl) phosphine hydrochloride (TCEP) (freshly prepared, stock solution: 50 mM; J60316-06, Fisher), and 1 µl TAMRA-N3 (stock solution: 1.25 mM in DMSO; T10182, Fisher), incubating at room temperature for 1 hr, and quenching the reaction by boiling with 4× sodium dodecyl sulfate (SDS) loading buffer. This was run on a Nupage 4–12% Bis-Tris protein gel (NP0323BOX, Invitrogen, Massachusetts, USA) and the in-gel fluorescence of the gel was visualized using a LI-COR Odyssey imaging system (LI-COR Biosciences, Nebraska, USA) at 600 nm for an exposure time of 30 s. After imaging the TAMRA signal, the same gel was coomassie stained (InstantBlue, ISB1L, Abcam, Cambridge, UK) and visualized the same way at 700 nm.

For proteomic identification of competed proteins, plasma samples were pooled from five different individuals and were crosslinked with either no photoprobe and equivalent amounts of DMSO, 5 µM photoprobe and DMSO, or 5 µM photoprobe and 100 µM PTX as described above. Each condition was set up as 24 individual reactions and pooled after crosslinking. To attach a biotin handle, 3 µl TBTA, 1 µl CuSO$_4$, 1 µl tris (2-carboxyethyl) phosphine hydrochloride (TCEP), and 1.14 µl Biotin-N3 (stock solution: 9.67 mM in DMSO; 1265, Click Chemistry Tools, Arizona, USA), were added for each reaction and this was incubated at room temperature for 1 hr, rotating. After incubation, each condition was run through a 3-kDa MWCO centrifuge filter twice (UFC800324, Amicon, Millipore-Sigma, Massachusetts, USA) to dilute excess Biotin-N3 until reaching a 900× dilution. Pulldown of biotinylated photoprobe–protein complexes was achieved with a magnetic bead strep pulldown following the protocol outlined in *Wei et al., 2021*. The pre and post-pulldown samples were run on a gel for a streptavidin blot (*Figure 2A*) and silver stain. After verifying pulldown efficacy, samples were run on SDS–polyacrylamide gel electrophoresis (PAGE) gels in two replicates (one lane each) for each condition. Gels were fixed in 50:50 water:MeOH with 10% acetic acid for 1–2 hr. For the first replicate, the gel was run for a short period, and an approximately one centimeter squared portion containing the whole lane for each condition was excised and fixed. For the second replicate, the gel was run completely and the proteins between 39 and 64 kDa were excised and fixed using the ladder as a size reference. Sections were chopped into 1 mm pieces under sterile conditions and stored at 4°C in 100 µl of water with 1% acetic acid until processed for proteomics.

## Proteomic identification of pulled down proteins across conditions

For proteomics analyses, SDS–PAGE gel slices approximately 1 cm in length were prepared for proteolytic digestion. Each fixed gel slice was diced into 1 mm cubes under sterile conditions, and then rinsed with 50 mM ammonium bicarbonate to remove residual acidification from the fixing process. Following rinsing, the gels were incubated in 80% acetonitrile in water for 5 min; the solvent was removed and then the gel pieces were incubated with 10 mM dithiothreitol (DTT) dissolved in water at room temperature for 20 min. Following reduction, alkylation was performed using 30 mM acrylamide for 30 min at room temperature to cap free reduced cysteines. Proteolysis was performed using trypsin/lysC (Promega, Wisconsin, USA) in 50 mM ammonium bicarbonate overnight at 37°C. Resulting samples were spun to pellet gel fragments prior to extraction of the peptides present in the supernatant. The resulting peptides were dried by speed vac before dissolution in a reconstitution buffer (2% acetonitrile with 0.1% formic acid), with an estimated 1 µg on-column used for subsequent LC–MS/MS analysis.

The LC–MS experiment was performed using an Orbitrap Eclipse Tribrid mass spectrometer RRID:022212 (Thermo Scientific, San Jose, CA) with liquid chromatography using an Acquity M-Class UPLC (Waters Corporation, Milford, MA). A flow rate of 300 nl/min was used, where mobile phase A was 0.2% formic acid in water and mobile phase B was 0.2% formic acid in acetonitrile. Analytical columns were prepared in-house with an I.D. of 100 µm pulled to a nanospray emitter using a P2000 laser puller (Sutter Instrument, California, USA). The column was packed using C18 reprosil Pur 1.8 µm stationary phase (Dr. Maisch, Entringen, Germany) to an approximate length of 25 cm. Peptides were

directly injected onto the analytical column using a 80-min gradient (2–45% B, followed by a high-B wash). The mass spectrometer was operated in a data-dependent fashion using CID fragmentation in the ion trap for MS/MS spectra generation.

For data analysis, the RAW data files were processed using Byonic v4.1.5 (Protein Metrics, California, USA) to identify peptides and infer proteins based on a proteomic reference created with the *O. sylvatica* genome (available on GenBank: JARQOD000000000) annotation. Proteolysis with Trypsin/LysC was assumed to be specific with up to two missed proteolytic cleavages. Precursor mass accuracies were held within 12 ppm, and 0.4 Da for MS/MS fragments in the ion trap. Cysteine modified with propionamide were set as fixed modifications in the search, and other common modifications (e.g. oxidation of methionine) were also included. Proteins were held to a false discovery rate of 1%, using standard reverse-decoy technique (*Elias and Gygi, 2007*).

## Identification of ABG proteins in different species and sequence confirmation

To identify potential ABG proteins in other species, we used the *Os*ABG protein sequence identified in the proteomics as the query and searched against blast databases created from the *A. femoralis* genome (available on GenBank: JARQOC000000000), and *E. tricolor*, *D. tinctorius* (*Alvarez-Buylla et al., 2022*), and *M. aurantiaca* transcriptomes. The top hit from each blast search was used as the most probable ABG gene from those species. To ensure that the sequences did not contain sequencing or alignment errors, the gene from *O. sylvatica, D. tinctorius, and E. tricolor* was amplified using PCR and sequence confirmed with sanger sequencing. Total RNA was extracted from flash frozen liver tissue from three lab-bred, non-toxic, individuals from each species using the Monarch total RNA Miniprep Kit (T2010S, New England Biolabs, Massachusetts, USA) following the manufacturer instructions. This was used to create cDNA using the SuperScript III First-Strand Synthesis kit (18080–400, Invitrogen), following manufacturer's instructions with an oligo(dT)20 primer. This was used for a PCR using Phusion High Fidelity DNA polymerase (F-530, Thermo Scientific, Massachusetts, USA) and the primers and cycling conditions described below. PCRs were analyzed using a 1% agarose gel for presence of a single band, cleaned up (NucleoSpin Gel and PCR cleanup, 740609.50, Takara Bio, Shiga, Japan), and transformed into pENTR vectors using a D-TOPO kit (45-0218, Invitrogen). Plasmids containing the ABG sequences from each individual were then mini prepped (27106X4, QIAGEN, Hilden, Germany) and sanger sequenced with M13F and M13R primers (Azenta Life Sciences, California, USA). Sequences were aligned using Benchling (Benchling Inc, California, USA) software, with the alignment methods MAFFT (*Katoh and Standley, 2013*) used for DNA alignments and Clustal Omega used for protein alignments.

| Species | fwd primer | rev primer | Tm (°C) |
|---|---|---|---|
| *O. sylvatica* | CACCATGAAACTTTTCGT CTACCTGTGTTTCAGC | CTATTTTGTTGGGTCT ACTATTCTTCCGCTG | 68 |
| *D. tinctorius* | CACCATGAAGCTTTTCGT CTTCCTATGTTTCAGCC | CTATTTTGTTGGGTTTAT TATTTTTCCATTCAAAATATCG | 66 |
| *E. tricolor* | CACCATGAAGCTTTTCA TCTTCCTGTGTTTGAGCC | CTATTTTGTTGGGTCT ATTATTCTTCCGGAGAAAAC | 68 |

Cycling conditions: 98°C for 30 s, [98°C for 10 s, Tm for 30 s, 72°C for 2 min] × 34 cycles, 72°C for 10 min.

## Protein structure prediction and molecular docking analyses

The *Os*ABG protein folding was predicted using the amino acid sequence, edited for point mutations found across all three individuals used for sequence verification, and the AlphaFold google colab notebook (*Jumper et al., 2021a*; *Jumper et al., 2021b*). The predicted structure is provided in the supplementary information. The default AlphaFold parameters were used. Molecular docking was performed using the UCSF Chimera software (*Pettersen et al., 2004*), using AutoDock Vina (*Trott and Olson, 2010*; *Eberhardt et al., 2021*) with the three dimensional structure of PTX **251D** (Pubchem CID 6440480). The whole protein was used as the search space with the default search parameters (5 binding modes, exhaustiveness of search of 8, and a maximum energy difference of 3 kcal/mol). The

docking result with the highest predicted affinity was used and is included in the supplementary files. Protein structures and docking were visualized using PyMol for publication quality images.

## Recombinant protein expression and binding assays

Recombinant ABG proteins were expressed by Kemp Proteins (Maryland, USA) through their custom insect cell protein expression and purification services. The reagents and vectors used are proprietary to Kemp Proteins, however the general expression and purification details are as follows. The verified protein sequences described above or the point mutations (*Figure 4*) were codon optimized for SF9 insect expression, and a 10× HIS tag was added to the C-terminal end. For *Os*ABG, a 1 liter expression was performed, for all other sequences (other species and mutants) a 50-ml expression was used. For the 1 liter expression, a multiplicity of infection of one was used for the p1 baculovirus and the supernatant was collected after 72 hr. To this, 5 ml of QIAGEN Ni-NTA resin washed and equilibrated in Buffer A (20 mM sodium phosphate, 300 mM NaCl, pH = 7.8) was added and it was mixed overnight at 4°C. Afterwards, it was packed in a 5-ml Bio-Scale column and washed with 3 column volumes (CV) of Buffer A, followed by washing with 5% Buffer B (20 mM sodium phosphate, 300 mM NaCl, 500 mM imidazole, pH = 7.8) for 5 CV. Protein was eluted with a linear gradient from 5 to 60% over 25 CV, and 6 ml fractions were collected throughout. All of the fractions containing protein were pooled and concentrated to 1 mg/ml using an Amicon centrifugal filter with a 10-kDa MWCO, the buffer was exchanged to PBS, it was filtered through a 0.2-µm filter, aliquoted, and frozen at −80°C. Protein expression and purification resulted in a clear band by western blot (*Figure 4—figure supplement 3*) and a clean doublet pattern by coomassie (*Figure 4—figure supplement 3*) closely resembling that seen in the plasma crosslinking results (*Figure 1C*) in both reduced and non-reduced conditions. For the 50-ml expression of *Dt*ABG, *Et*ABG, and mutant *Os*ABG proteins, a 10% ratio of p1 virus to media was used and the supernatant was collected after 72 hr, to which 1 ml of QIAGEN Ni-NTA resin washed and equilibrated in Buffer A (20 mM sodium phosphate, 300 mM NaCl, pH = 7.4) was added. This was mixed overnight at 4°C and then packed into a 1-ml Bio-Scale column, washed with 3 CV of Buffer A, washed with 5 CV of 5% Buffer B (20 mM sodium phosphate, 300 mM NaCl, 500 mM imidazole, pH = 7.4), and eluted with 5 CV of 50% Buffer B. Fractions containing protein were buffer exchanged into PBS, and the final concentrations were approximately 0.2 mg/ml, with varying final volumes. Protein expression and purification resulted in a clear band by western blot (*Figure 4—figure supplement 3*), and a clean doublet by coomassie (*Figure 4—figure supplement 3*) in both reduced and non-reduced conditions.

## Determination of dissociation constant using MST

To determine the binding affinity of *Os*ABG for PTX, we used MST using the Monolith system (Nanotemper Technologies, München, Germany). Purified wild-type *Os*ABG, *Os*ABG mutant 3 (D383A), and BSA protein were labeled using the protein labeling kit Red-NHS 2nd generation (MO-L011, Nanotemper) which dyes primary lysine residues in the protein. The kit was used following the manufacturer's instructions, however a 1.5× excess of dye was used instead of 3× as this was found to better achieve a degree of labeling of ~0.5. To remove aggregates during the assay, PBS-Tween was used for protein labeling and all dilutions. The labeled protein was centrifuged for 10 min at 20,000 × $g$ on a benchtop centrifuge, and the supernatant was taken to further remove any aggregation. The concentration was measured prior to calculating and setting up dilution series. A final concentration of 5 nM *Os*ABG was used, and a 16 tube 2× serial dilution series of PTX **251D** was made with the highest concentration being 5000 µM. The concentration of DMSO was maintained consistent across the dilution series. The labeled *Os*ABG was incubated for 10–30 min prior to loading into capillaries, and three biological replicates were pipetted and run separately. The Monolith premium capillaries (MO-K025, Nanotemper) were used, the MST power was set to Medium, and the excitation power was set to auto-detect. The three replicates were compiled and plotted together using GraphPad Prism (GraphPad Software, California, USA), and a dissociation model was fit to the data. The raw data are included in the supplementary information.

## Determination of free versus bound alkaloids

Solutions with 4 µM of *Os*ABG protein, 4 µM of either PTX **251D** or nicotine, and a final volume of 100 µl were made and incubated for 1 hr at room temperature. This was transferred to a 3-kDa MWCO

centrifugal filter and spun at max speed on a benchtop centrifuge at 4°C for 45 min. The top and bottom fractions were brought up to 100 µl with ultrapure water and transferred to new tubes, where 300 µl of 2:1 acetonitrile:methanol was added, after which they were vortexed and centrifuged at max speed on a benchtop centrifuge at 4°C for 10 min. The supernatant was transferred to autosampler vials for quantitation of the amount of alkaloid in each fraction with mass spectrometry. Each condition was run in triplicate. Samples were analyzed using an Agilent Quadrupole time-of-flight LC–MS instrument (Agilent Technologies, California, USA), with MS analysis performed by electrospray ionization in positive mode. Metabolites were separated with an Eclipse Plus C18 column (959961-902, Agilent) with normal phase chromatography. Mobile phases were: Buffer A (water with 0.1% formic acid) and Buffer B (90% acetonitrile, 10% water with 0.1% formic acid). The flow rate was maintained constant at 0.7 ml/min throughout the LC protocol. The LC gradient elution was set as follows: starting at 5% B held till 0.51 min, linear gradient from 5 to 25% B in 1.5 min, linear gradient from 25 to 50% B in 23 min, linear gradient from 50 to 95% B in 30 s, 95% B held for 2 min, linear gradient from 95 to 5% B in 1 min, and 5% B held for 1.5 min to equilibrate the column to the initial conditions. The total run time was 30 min and the injection volume was 10 µl. Data were analyzed using the Agilent MassHunter software; the extracted ion chromatograms for PTX were searched using the exact mass $M + 1$ of 252.2333, and nicotine was searched using the exact mass $M + 1$ of 163.123, with a tolerance of a symmetric ± 100 ppm. Extracted ion chromatograms were smoothed once before automatically integrating to get the abundance values. Abundance values were used to calculate the fractions above and below the filter for each replicate, which were then plotted with GraphPad. All raw data are provided as mzXML files through DataDryad: https://doi.org/10.5061/dryad.mkkwh7143.

## Field collections of *O. sylvatica*

The frog samples used in this paper are the same as those used for the project described in **Moskowitz et al., 2022**. For each location, 10 *O. sylvatica* individuals were collected under collection permit 0013-18 IC-FAU-DNB/MA issued by the Ministerio del Ambiente de Ecuador, between the hours of 7:00 and 18:00 during early May to early June 2019. All individuals were euthanized the same day as collection. Prior to euthanizing, frogs were sexed, weighed, and the snout-vent length was measured. Orajel (10% benzocaine) was used as an anesthetic prior to cervical dislocation. Once euthanized, frogs were immediately dissected and the liver, intestines, and half of the dorsal skin were stored in RNAlater in cryotubes at room temperature. The other half of the dorsal skin was placed in methanol in glass tubes at room temperature. Once back in the lab, all tissues were stored at −20°C until further processing. All tissues were transported to the United States under CITES permits 19EC000036/VS, 19EC000037/VS, and 19EC000038/VS.

## Alkaloid extraction, detection, and analysis

All following steps were performed under a hood. Skins were taken out of methanol with forceps and weighed. From the methanol that the skin was stored in, 1 ml was taken and syringe filtered through a 0.45-µ PTFE filter (44504-NP, Thermo Scientific) into the new glass vial with a PTFE cap (60940A-2, Fisher) filled with 25 µl of 1 µg/µl (−)-Nicotine (N3876-100ML, Sigma-Aldrich), for a total of 25 µg of added nicotine. Tubes were capped and vortexed, and stored at −80°C for 24 hr, during which proteins and lipids should precipitate. After 24 hr, tubes were taken out of the −80°C and quickly syringe filtered through a 0.45-µ PTFE filter again into a new glass vial. A 100-µl aliquot was added to an GC–MS autosampler vial, and the remaining solution was stored in the original capped vial at −80°C.

GC–MS analysis was performed on a Shimadzu GCMS-QP2020 instrument with a Shimadzu 30 m × 0.25 mm ID SH-Rxi-5Sil MS column closely following the protocol outlined in **Saporito et al., 2010**. In brief, GC separation of alkaloids was achieved using a temperature program from 100 to 280°C at a rate of 10°C/min with He as the carrier gas (flow rate: 1 ml/min). This was followed by a 2-min hold and additional ramp to 320°C at a rate of 10°C/min for column protection reasons, and no alkaloids appeared during this part of the method. Compounds were analyzed with electron impact-mass spectrometry. The GC–MS data files were exported as CDF files and the Global Natural Products Social Network (GNPS) was used to perform the deconvolution and library searching against the AMDIS (NIST) database to identify all compounds (https://gnps.ucsd.edu; **Wang et al., 2016**). For deconvolution (identification of peaks and abundance estimates) the default parameters were used, for the

library search the precursor ion mass tolerance was set to 20,000 Da and the MS/MS fragment ion tolerance to 0.5 Da. The resulting dataset was filtered to keep only compounds that matched to our spiked-in nicotine standard, alkaloids previously found in poison frogs from the Daly 2005 database (*Daly et al., 2005*), or compounds with the same base ring structure and R groups as the classes defined in *Daly et al., 2005*. All GC–MS data as CDF files are available through DataDryad: https://doi.org/10.5061/dryad.mkkwh7143.

Once the feature table from the GNPS deconvolution was filtered to only include only poison frog alkaloids and nicotine, the abundances values were normalized by dividing by the nicotine standard and skin weight. This filtered and normalized feature table was used for all further analyses and visualizations. All steps were carried out with R version 4.0.4, and code is included in Supplementary file corresponding to *Figure 6*: PlottingExpressionAndToxicity.R.

## RNA extraction and library preparation

RNA extraction followed the Trizol (15596018, Thermo Fisher) RNA isolation protocol outlined in *Caty et al., 2019* according to the manufacturer's instructions, and with sample randomization to avoid batch effects. RNA quality was measured on a Agilent Tapestation RNA screentape (Agilent), and quantified using a Qubit Broad Range RNA kit (Q10210, Invitrogen). In the liver and intestines, samples with RIN scores greater than 5 were kept, RNA was normalized to the same Qubit concentration, and mRNA was isolated and library prepped using the NEB Directional RNA sequencing kit (E7765L, New England Biolabs) with the PolyA purification bundle (E7490L, New England Biolabs) and 96 Unique Dual Indices (E7765L, New England Biolabs). The skin RIN scores were much lower, signaling potential RNA degradation, ribosomal degradation was instead used to isolate mRNA. Following normalization within all skin RNA samples to the same Qubit concentration, we used the Zymo RiboFree Total RNA Library Prep kit (R3003-B, Zymo Research, California, USA) following the manufacturer's instructions. After library prep for all tissues was complete library size was quantified with the Agilent Tapestation D1000 screentape, and concentration was measured with the Qubit dsDNA high sensitivity kit (Q33231, Invitrogen). All libraries within a tissue type were pooled to equimolar amounts and sequenced on two lanes of an Illumina NovaSeq (Illumina, California, USA) machine to obtain 150 bp paired-end reads.

## RNA expression analysis and identification of *O. sylvatica* serpinA genes

Analysis of RNA expression levels followed the protocol outlined by *Payne et al., 2022*. The Trim-galore! wrapper tool (*Krueger et al., 2021*) was used to trim adapter sequences with cutadapt (*Martin, 2011*) and quality filter the reads (trim_galore `--paired --phred33 --length` 36 -q 30 `--stringency` 1 -e 0.001). All trimmed reads are available through the NCBI BioProject PRJNA909817. *Kallisto* (*Bray et al., 2016*) was used to pseudoalign the reads to a reference created with the coding sequence of the annotated *O. sylvatica* genome (GenBank JARQOD000000000). These abundances were combined into a matrix, and the trimmed-mean of *M*-values (TMM) normalized counts were used for all further analyses. Additional serpinA genes were found in the genome by searching for all genes annotated with 'serpina' in the header, and by blasting the *Os*ABG protein sequence against the genome (*e*-value <1*e*−60) and including any additional genes not annotated with 'serpina'. Four sequences were removed because they were either exact matches (OopSylGTT00000004683), the N-terminal end (OopSylGTT00000004650, OopSylGTT00000004685), or the C-terminal (OopSylGTT00000004676) end of another serpina gene, and therefore could be potential duplications caused by annotation or assembly errors. To create the protein tree (*Figure 6B*), ClustalW was used to align the sequences, a distance matrix was created using identity, and neighbor joining was used to construct the tree. The albumin gene was determined by blasting the protein sequences of *Xenopus laevis* albumin A (Uniprot #P08759), *X. laevis* albumin B (Uniprot #P14872), and the Asian toad *Bombina maxima* albumin (Uniprot #Q3T478) against the *O. sylvatica* genome. In all three cases, the top hit was the same (OopSylGTT00000003067), therefore this was assumed to be the most likely albumin candidate in the genome and was used to plot the TMM expression for comparison. All plots were created in R version 4.0.4, and all analysis and plotting code are available in the Supplementary file corresponding *Figure 6*: PlottingExpressionAndToxicity.R.

## Custom *Os*ABG antibody blotting and staining

We obtained a custom rabbit anti-*Os*ABG antibody from Pocono rabbit farm using purified *Os*ABG protein as the antigen. We performed an antibody blocking assay to assess the specificity of the staining obtained with the custom-designed *Os*ABG antibody. For western blotting, the following primary antibody solutions were incubated for 3 hr at room temperature on a nutator prior to blotting: (1) 1:1000 dilution of the anti-*Os*ABG primary antibody and (2) 1:1000 anti-*Os*ABG primary antibody with a fivefold excess of purified *Os*ABG protein by volume (1:200 dilution of 1 mg/ml protein stock). The same amount of denatured protein or lysate stock was used for each well, the gel for both conditions was run and transferred together, and horse-radish peroxidase substrate (1721064, BioRad Laboratories, California, USA) was used to visualize the banding pattern. For immunohistochemistry, we pre-incubating a 1:800 dilution of the *Os*ABG antibody with a fivefold excess (1:160) of the purified *Os*ABG protein (1 mg/ml) at 4°C on a nutator for 12 hr. The negative control with no antibody, and positive control with only a 1:800 dilution of the anti-*Os*ABG antibody and no protein were also incubated at 4°C on a nutator for 12 hr. Right before the incubation of the tissue sections, mouse monoclonal antibody against actin (AC-40, ab11003, Abcam) was added to the pre-incubated mix in a 1:800 dilution. The detailed tissue sectioning, tissue preparation, staining, and imaging steps are described below.

For visualizing the tissue distribution of *Os*ABG, intestines, liver, muscle, and skin tissue from one adult *O. sylvatica* were obtained, cut into small pieces and fixed overnight in 4% paraformaldehyde in PBS buffer (pH 7.4) at 4°C. The next day, tissues were rinsed three times for 15 min in PBS and immersed in sterile filtered 30% sucrose in PBS until the tissue sank to the bottom of the tube (max 48 hr). After this, the tissues were embedded in Tissue-Tek OCT (Sakura Finetek, California, USA), frozen on dry ice, and stored at −80°C for subsequent cryosectioning on a Leica CM1860 cryostat (Leica Biosystems, Illinois, USA). Tissue was sectioned into 17 µm and mounted to Superfrost slides (48311-703, VWR International, Pennsylvania, USA) such that each slide contained each tissue type in duplicate. Slides were dried at room temperature for 2 days before being processed for immunofluorescence. Sections on each slide were surrounded with a hydrophobic PAP Pen (H-4000, Vector Laboratories, California, USA) and rehydrated with Tris-buffered saline (TBS, pH 7.4) three times for 15 min. Rehydrated sections were incubated with a blocking solution of TBS containing 0.3% Triton X-100 and 10% normal goat serum (NGS) for 2 hr at room temperature. All incubation steps were performed flat on a grid in a hybridization chamber containing tissues soaked in sterile water to prevent sections from drying. After blocking, slides were either incubated in 300 µl of (1) TBS containing 10% NGS and no antibodies (negative control), (2) TBS containing 10% NGS with rabbit anti-*Os*ABG antibody and mouse anti-actin antibody both at a 1:800 dilution, or (3) TBS containing 10% NGS with rabbit anti-*Os*ABG antibody (1:800 dilution) pre-incubated with a fivefold excess by volume (1:160 dilution) of purified *Os*ABG protein (at a stock solution of 1 mg/ml) and mouse anti-actin antibody (1:800 dilution). Incubation with the primary antibody was performed overnight at 4°C after which slides were rinsed with TBS three times for 15 min each. All slides were then incubated for 2 hr at room temperature covered from light with a mix of goat anti-rabbit Alexa Fluor 568 (A-11011, Invitrogen) to visualize *Os*ABG and goat anti-mouse Alexa Fluor 488 (A-11001, Invitrogen) to visualize actin both at a 1:400 dilution in TBS containing 0.3% Triton X-100. Slides were rinsed three times for 15 min in TBS, followed by a rinse in deionized water before mounting them with FluoShield aqueous mounting media containing DAPI (ab104139, Abcam) and drying them overnight at 4°C. Staining was imaged on a Leica DM6000 fluorescent microscope at a 20× magnification, pictures were taken with a K9 monochromatic camera with exposure times of 80ms for *Os*ABG (Alexa 568), 40ms for actin (Alexa 488) and between 2 and 20 ms for DAPI (359 nm) depending on the tissue. Figures were compiled using the freeware Inkscape and no manipulations other than cropping were used on the images.

## Acknowledgements

The authors acknowledge that this research was conducted on the ancestral lands of the Muwekma Ohlone people at Stanford. We understand the implications of the historical and present colonialism the Ohlone people experience and celebrate their continued stewardship of their lands. We thank María Dolores Guarderas and Andrea Terán Valdéz for their assistance coordinating field work and their kindness. We thank Ariel Rodríguez for assistance with the *O. sylvatica* genome annotation. Thank you to the Mabel Gonzalez and the Dorrestein lab for their assistance using the GNPS tool to identify

alkaloids in our samples. We would also like to thank the Laboratory of Organismal Biology, the Long Lab, the Soh lab, Joel Francis, and Cheyenne Payne for helpful discussions and guidance throughout this project. Centro Jambatu researchers thank Wikiri and Saint Louis Zoo for their commitment and sustained support for amphibian research.

## Additional information

### Funding

| Funder | Grant reference number | Author |
|---|---|---|
| National Science Foundation | IOS-1822025 | Lauren A O'Connell |
| New York Stem Cell Foundation | | Lauren A O'Connell |
| National Science Foundation Graduate Research Fellowship Program | DGE-1656518 | Aurora Alvarez-Buylla |
| Howard Hughes Medical Institute | GT13330 | Aurora Alvarez-Buylla |
| Fundacion Alfonso Martin Escudero | | Maria Dolores Moya Garzon |
| Wu Tsai Human Performance Alliance | | Maria Dolores Moya Garzon |

The funders had no role in study design, data collection, and interpretation, or the decision to submit the work for publication.

### Author contributions

Aurora Alvarez-Buylla, Conceptualization, Resources, Data curation, Software, Formal analysis, Validation, Investigation, Visualization, Methodology, Writing – original draft, Project administration, Writing – review and editing; Marie-Therese Fischer, Formal analysis, Validation, Investigation, Visualization, Methodology; Maria Dolores Moya Garzon, Alexandra E Rangel, Formal analysis, Validation, Investigation, Methodology; Elicio E Tapia, Resources, Validation, Investigation, Methodology; Julia T Tanzo, Methodology; H Tom Soh, Luis A Coloma, Resources, Methodology; Jonathan Z Long, Conceptualization, Resources, Supervision, Methodology, Writing – review and editing; Lauren A O'Connell, Conceptualization, Resources, Supervision, Funding acquisition, Writing – review and editing

### Author ORCIDs

Aurora Alvarez-Buylla https://orcid.org/0000-0001-6256-0300
Jonathan Z Long https://orcid.org/0000-0003-2631-7463
Lauren A O'Connell https://orcid.org/0000-0002-2706-4077

### Ethics

All animal procedures were approved by the Institutional Animal Care and Use Committee at Stanford (protocol #34153). Topical benzocaine was used for anesthesia prior to the euthanasia of all animals. Laboratory-bred animals were purchased from Understory Enterprises (Ontario, Canada) or Josh's Frogs (Michigan, USA) depending on the species. Animals were either euthanized for plasma collection upon arrival, or housed in 18x18x18 inch glass terraria, and fed a diet of non-toxic Drosophila melanogaster until euthanasia. Plasma and tissues from a total of 62 animals were used for this study, consisting of 32 lab-bred animals and 30 field-collected animals that are described below. Sample size of field-collected animals was determined based on variability seen in previous studies, and for laboratory experiments based on experimental needs in terms of volume of plasma.

### Decision letter and Author response

Decision letter https://doi.org/10.7554/eLife.85096.sa1
Author response https://doi.org/10.7554/eLife.85096.sa2

## Additional files

### Supplementary files
• MDAR checklist

### Data availability

All raw data, analysis scripts, and intermediate data analysis are available either as Source Data zip files for each figure, or through public repositories. Uncropped gel images, analysis and plotting code, raw and normalized MST data, normalized RNAseq data, and other intermediate analysis files used to make figures are available as 'Source Data' zip files included with the submission. All raw proteomics and mass spectrometry data are available on Dryad. The official ABG sequences can be found through GenBank with the following accessions: *O. sylvatica* (OQ032869), *D. tinctorius* (OQ032870), and *E. tricolor* (OQ032871). The *O. sylvatica* genome, field collected sample information, raw and trimmed sequencing reads are available through the NCBI BioProject PRJNA909817. The *A. femoralis* genome and raw sequencing data is available through NCBI BioProject PRJNA913987. Annotated versions of the *A. femoralis* and *O. sylvatica* genomes, as well as *E. tricolor*, *D. tinctorius*, and *M. aurantiaca* transcriptomes are available on Dryad.

The following datasets were generated:

| Author(s) | Year | Dataset title | Dataset URL | Database and Identifier |
|---|---|---|---|---|
| Alvarez-Buylla A, O'Connell LA | 2023 | Binding and sequestration of poison frog alkaloids by a plasma globulin | https://doi.org/10.5061/dryad.mkkwh7143 | Dryad Digital Repository, 10.5061/dryad.mkkwh7143 |
| Alvarez-Buylla A, Moya Garzon MD, Rangel AE, Tapia EE, Soh HT, Tanzo J, Coloma LA, Long JZ, O'Connell LA | 2023 | Oophaga sylvatica genome sequencing, assembly, and gene expression | https://www.ncbi.nlm.nih.gov/bioproject/PRJNA909817 | NCBI BioProject, PRJNA909817 |
| Alvarez-Buylla A, Moya-Garzon MD, Rangel AE, Tapia E, Tanzo J, Soh HT, Coloma LA, Long JZ, O'Connell LA | 2023 | Allobates femoralis genome sequencing and assembly | https://www.ncbi.nlm.nih.gov/bioproject/PRJNA913987 | NCBI BioProject, PRJNA913987 |

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

# Appendix 1

## Appendix 1—key resources table

| Reagent type (species) or resource | Designation | Source or reference | Identifiers | Additional information |
|---|---|---|---|---|
| Gene (*Oophaga sylvatica*) | *Os*ABG | This paper | GenBank: OQ032869 | Consensus sequence from three individuals |
| Gene (*Dendrobates tinctorius*) | *Dt*ABG | This paper | GenBank: OQ032870 | Consensus sequence from three individuals |
| Gene (*Epipedobates tricolor*) | *Et*ABG | This paper | GenBank: OQ032871 | Consensus sequence from three individuals |
| Strain, strain background (*Escherichia coli*) | TOP10 | Invitrogen | Cat# C404010 | Chemically competent cells |
| Genetic reagent (*Oophaga sylvatica*) | *O. sylvatica* genome | This paper | GenBank: JARQOD000000000 | Annotation available with associated datadryad files |
| Genetic reagent (*Allobates femoralis*) | *A. femoralis* genome | This paper | GenBank: JARQOC000000000 | Annotation available with associated datadryad files |
| Genetic reagent (*Epipdobates tricolor*) | *E. tricolor* transcriptome | This paper | | Available with datadryad files |
| Genetic reagent (*Dendrobates tinctorius*) | *D. tinctorius* transcriptome | *Alvarez-Buylla et al., 2022* | PMID: 35275922 | |
| Genetic reagent (*Mantella aurantiaca*) | *M. aurantiaca* transcriptome | This paper | | Available with datadryad files |
| Cell line (*Spodoptera frugiperda*) | Sf9 insect cell culture | Kemp Proteins | | Proprietary insect cell expression technology |
| Biological sample (*Oophaga sylvatica*) | Captive-bred little devil poison frogs | Understory Enterprises | | |
| Biological sample (*Dendrobates tinctorius*) | Captive-bred dyeing poison frogs | Josh's Frogs | | |
| Biological sample (*Epipedobates tricolor*) | Captive-bred phantasmal poison frogs | Josh's Frogs | | |
| Biological sample (*Allobates femoralis*) | Captive-bred brilliant-thighed poison frogs | Understory Enterprises | | |
| Biological sample (*Mantella aurantiaca*) | Captive-bred golden mantella | Josh's Frogs | | |
| Biological sample (*Homo sapiens*) | Human plasma | Innovative Research | Cat# IPLANAH | |
| Biological sample (*Oophaga sylvatica*) | Field-collected little devil poison frogs | *Moskowitz et al., 2022* | doi:10.1101/2022.06.14.495949 | Tissue from 30 individuals |
| Antibody | Anti-*Os*ABG rabbit polyclonal antibody | This paper | | Generated by Pocono rabbit farm, WB dilution 1:1000, IHC dilution 1:800 |
| Antibody | Anti-actin mouse monoclonal antibody | Abcam | Cat# ab11003 | IHC dilution 1:800 |
| Antibody | Goat monoclonal anti-rabbit Alexa Fluor 568 | Invitrogen | Cat# A-11011 | IHC dilution 1:400 |
| Antibody | Goat monoclonal anti-mouse Alexa Fluor 488 | Invitrogen | Cat# A-11001 | IHC dilution 1:400 |
| Recombinant DNA reagent | pENTR plasmid | Invitrogen | Cat# K240020 | D-TOPO cloning |
| Sequence-based reagent | *Os*ABG_fwd | This paper | PCR primer | CACCATGAAACTTTTCG TCTACCTGTGTTTCAGC |
| Sequence-based reagent | *Os*ABG_rev | This paper | PCR primer | CTATTTTGTTGGGTCTA CTATTCTTCCGCTG |

*Appendix 1 Continued on next page*

*Appendix 1 Continued*

| Reagent type (species) or resource | Designation | Source or reference | Identifiers | Additional information |
|---|---|---|---|---|
| Sequence-based reagent | *Dt*ABG_fwd | This paper | PCR primer | CACCATGAAGCTTTTCGT CTTCCTATGTTTCAGCC |
| Sequence-based reagent | *Dt*ABG_rev | This paper | PCR primer | CTATTTTGTTGGGTTTATT ATTTTTCCATTCAAAATATCG |
| Sequence-based reagent | *Et*ABG_fwd | This paper | PCR primer | CACCATGAAGCTTTTCAT CTTCCTGTGTTTGAGCC |
| Sequence-based reagent | *Et*ABG_rev | This paper | PCR primer | CTATTTTGTTGGGTCTA TTATTCTTCCGGAGAAAAC |
| Peptide, recombinant protein | OsABG | This paper | GenBank: OQ032869 | Custom expression and purification by Kemp Proteins in sf9 cell culture |
| Peptide, recombinant protein | OsABG mutant 1 | This paper | Y36A + W276A + S374A + D383A | Custom expression and purification by Kemp Proteins in sf9 cell culture |
| Peptide, recombinant protein | OsABG mutant 2 | This paper | Y36A + S268A + D273A + D383A | Custom expression and purification by Kemp Proteins in sf9 cell culture |
| Peptide, recombinant protein | OsABG mutant 3 | This paper | D383A | Custom expression and purification by Kemp Proteins in sf9 cell culture |
| Peptide, recombinant protein | OsABG mutant 4 | This paper | Y36F | Custom expression and purification by Kemp Proteins in sf9 cell culture |
| Peptide, recombinant protein | OsABG mutant 5 | This paper | S374A | Custom expression and purification by Kemp Proteins in sf9 cell culture |
| Peptide, recombinant protein | DtABG | This paper | GenBank: OQ032870 | Custom expression and purification by Kemp Proteins in sf9 cell culture |
| Peptide, recombinant protein | EtABG | This paper | GenBank: OQ032871 | Custom expression and purification by Kemp Proteins in sf9 cell culture |
| Peptide, recombinant protein | Bovine serum albumin (BSA) | Sigma-Aldrich | Cat# A2153-50G | |
| Commercial assay or kit | Monarch total RNA Miniprep Kit | NEB | Cat# T2010S | |
| Commercial assay or kit | Superstrand III First-Strand Synthesis kit | Invitrogen | Cat# 18080-400 | ligo(dT)20 primer used |
| Commercial assay or kit | Phusion High Fidelity DNA polymerase | Thermo Scientific | Cat# F-530 | |
| Commercial assay or kit | NucleoSpin Gel and PCR cleanup | Takara Bio | Cat# 740609.50 | |
| Commercial assay or kit | pENTR/D-TOPO kit | Invitrogen | Cat# 45-0218 | |
| Commercial assay or kit | Miniprep Kit | QIAGEN | Cat# 27106X4 | |
| Commercial assay or kit | Sanger Sequencing | Azenta Life Sciences | | M13F and M13R primers used |
| Commercial assay or kit | Red-NHS 2nd Generation Labeling Kit | Nanotemper | Cat# MO-L011 | |
| Commercial assay or kit | Tapestation RNA screentape analysis | Agilent | Cat# 5067–5576; Cat# 5067–5578; Cat# 5067–5577 | |
| Commercial assay or kit | Qubit Broad Range RNA kit | Invitrogen | Cat# Q10210 | |
| Commercial assay or kit | NEB Directional RNA sequencing Kit | NEB | Cat# E7765L | |
| Commercial assay or kit | Zymo RiboFree TotalRNA Library Prep Kit | Zymo Research | Cat# R3003-B | |
| Commercial assay or kit | Tapestation D1000 screentape analysis | Agilent | Cat# 5582; Cat# 5583 | |
| Commercial assay or kit | Qubit dsDNA high sensitivity kit | Invitrogen | Cat# Q33231 | |

*Appendix 1 Continued on next page*

*Appendix 1 Continued*

| Reagent type (species) or resource | Designation | Source or reference | Identifiers | Additional information |
|---|---|---|---|---|
| Commercial assay or kit | Horse-Radish Peroxidase (HRP) substrate kit | Bio-Rad | Cat# 1721064 | |
| Chemical compound, drug | 'PTX 251D; pumiliotoxin 251D; PTX' | Other | Pubchem_CID:6440480 | Custom-synthesized molecule produced by PepTech (Burlington, MA, USA) |
| Chemical compound, drug | 'decahydroquinoline; DHQ' | Sigma-Aldrich | Cat# 125741 | |
| Chemical compound, drug | 'epibatidine; epi' | Sigma-Aldrich | Cat# E1145 | |
| Chemical compound, drug | 'histrionicotoxin-like compound; HTX' | Sigma-Aldrich | Cat# ENAH2C55884A-50MG | |
| Chemical compound, drug | 'indolizidine; indol' | Sigma-Aldrich | Cat# ATE24584802-100MG | |
| Chemical compound, drug | 'nicotine; nic' | Sigma-Aldrich | Cat# N3876-100ML | |
| Chemical compound, drug | 'cortisol; cort' | Sigma-Aldrich | Cat# H0888-1G | |
| Chemical compound, drug | 'photoprobe, PB' | Enamine | Cat# Z2866906198 | |
| Chemical compound, drug | TBTA | Fisher | Cat# H66485-03 | |
| Chemical compound, drug | Copper (II) sulfate | Fisher | Cat# BP346-500 | |
| Chemical compound, drug | Tris (2-carboxyethyl) phosphine hydrochloride | Fisher | Cat# J60316-06 | |
| Chemical compound, drug | TAMRA-N3 | Fisher | Cat# T10182 | |
| Chemical compound, drug | InstantBlue | Abcam | Cat# ISB1L | |
| Chemical compound, drug | Biotin-N3 | Click Chemistry Tools | Cat# 1265 | |
| Chemical compound, drug | Trizol | Thermo Fisher | Cat# 15596018 | |
| Chemical compound, drug | Tissue-Tek OCT | Sakura Finetek | Cat# 4583 | |
| Chemical compound, drug | FluoShield aqueous mounting media containing DAPI | Abcam | Cat# ab104139 | |
| Software, algorithm | Byronic | Protein Metrics | | v4.1.5 |
| Software, algorithm | MAFFT nucleotide sequence alignment | Benchling | | |
| Software, algorithm | Clustal Omega AA sequence alignment | Benchling | | |
| Software, algorithm | AlphaFold | *Jumper et al., 2021b* | PMID: 34265844 | Through google collab notebook: https://colab.research.google.com/github/deepmind/alphafold/blob/main/notebooks/AlphaFold.ipynb |
| Software, algorithm | UCSF Chimera | *Pettersen et al., 2004* | PMID: 15264254 | |
| Software, algorithm | Autodock Vina | *Eberhardt et al., 2021* | PMID: 34278794 | v1.2.0 |
| Software, algorithm | GraphPad Prism | GraphPad Software | | |

*Appendix 1 Continued on next page*

*Appendix 1 Continued*

| Reagent type (species) or resource | Designation | Source or reference | Identifiers | Additional information |
|---|---|---|---|---|
| Software, algorithm | GNPS mass spec deconvolution + identification | *Wang et al., 2016* | PMID: 27504778 | |
| Software, algorithm | R | CRAN | | v4.0.4 |
| Software, algorithm | Trim-galore! | *Martin, 2011* | doi:https://doi.org/10.14806/ej.17.1.200 | trim_galore --paired --phred33 --length 36 -q 30 --stringency 1 -e 0.001 |
| Software, algorithm | Kallisto | *Bray et al., 2016* | PMID: 27043002 | |
| Software, algorithm | ClustalW | *Thompson et al., 1994* | PMID: 7984417 | |
| Other | Nupage 4–12% Bis-Tris protein gel | Invitrogen | Cat# NP0323BOX | Pre-cast protein gels |
| Other | 3 kDa Amicon MWCO centrifuge filter | Millipore-Sigma | Cat# UFC800324 | Centrifugal molecular weight cutoff filters |
| Other | Monolith Premium Capillaries | Nanotemper | Cat# MO-K025 | Glass capillaries for microscale thermophoresis measurements |
| Other | Eclipse Plus C18 column | Agilent | Cat# 959961-902 | Chromatographic column for LC–MS/MS |
| Other | PTFE syringe filter | Thermo Scientific | Cat# 44504-NP | Consumable to filter samples prior to GC–MS analysis |
| Other | Glass vials with PTFE-lined caps | Fisher | Cat# 60940A-2 | Consumable to store samples prior to GC–MS analysis |
| Other | Superfrost Slides | VWR | Cat# 48311-703 | Slides used for IHC staining |
| Other | Hydrophilic PAP Pen | Vector laboratories | Cat# H-4000 | Hydrophilic barrier pen used in IHC staining protocol |

