## [Editor Report]

Poison frogs contain alkaloids in their plasma that make them toxic or unpalatable to predators, but how these animals avoid damage to themselves from their own defenses is not well understood. This valuable study identifies an alkaloid-binding protein, a member of the serpin superfamily, in the plasma of poison frogs that may explain how these animals are able to sequester a diverse array of alkaloids. With a convincing series of experiments, the authors advance our knowledge of the roles of serpins in animal ecophysiology.

---

## [Decision Letter]

**Decision letter after peer review:**

Thank you for submitting your article "Binding and sequestration of poison frog alkaloids by a plasma globulin" for consideration by *eLife*. Your article has been reviewed by 2 peer reviewers, one of whom is a member of our Board of Reviewing Editors, and the evaluation has been overseen by Meredith Schuman as the Senior Editor. The reviewers have opted to remain anonymous.

Overall, the reviewers were positive about this work. The reviewers have discussed their reviews with one another, and the Reviewing Editor has drafted this to help you prepare a revised submission.

Essential revisions:

1) Since the photoprobe differs significantly from PTX, it would be good to analyze the binding pocket mutations identified in Figure 4 using the direct binding assay of PTX in Figure 5A. Either the mutants or some generic albumin would be a good and essential control for this figure to establish the specificity of ABG.

2) Although the protein gels in Figure 1-2 show clearly the role of ABG, a ~50 kDa protein, it's unclear whether transferrin-like proteins, which are ~80 kDa, may also play a role because the gels show proteins between 39-64 kDa (Figure 1). The gel in Figure 2A is specific to one O. sylvatica and extends this range but the gel does not appear to be labeled accordingly, making it unclear whether other larger proteins could have been detected in addition to ABG. Related to this point, did the proteomic analysis (Figure 2B and elsewhere) reveal any hint of albumin? Is it surprising that albumin is not a strong hit, especially in light of the earlier literature implicating these proteins as sequesterers of the alkaloids? Clarifying these issues would facilitate the interpretation of the results.

3) There is what seems to be a significant size difference between the O. sylvatica bands and bands from the other toxic frog species, namely D. tinctorius and E. tricolor. Could the photoprobe be binding to other non-ABG proteins of different sizes in different frog species? Given that O. sylvatica bands are bright and this species was the only one subject to proteomics quantification, a possible conclusion may be that the ABG toxin sponge is a lineage-specific adaptation of O. sylvatica rather than a common mechanism of toxin sequestration among multiple independent lineages of poison frogs. It would be helpful if the authors could address this observation of their binding data and the hypothesis flowing from that in the manuscript.

4) Figure 4B-C: Photoprobe binding results in the presence of epi and nicotine appear to be missing for D. tinctorius and those in the presence of PTX and nicotine are missing for D. tricolor. Adding these results would make for a more complete picture of alkaloid binding by ABG in non-O. sylvatica species.

5) Using recombinant proteins with mutations at residues forming the binding pocket of O. sylvatica ABG (as inferred from docking simulations), the authors found that all binding pocket mutations disrupted photoprobe binding completely in vitro (L221-222, Figure 4E). However, there is no information presented on non-binding pocket mutations. Mutations outside of the binding pocket would presumably maintain photoprobe binding – barring any indirect structural changes that might disrupt binding pocket interactions with the photoprobe. This result is important for the conclusion that the binding pocket itself is the sole mediator of toxin interactions. The authors do show that one binding pocket mutation (D383A) results in some degree of photoprobe binding (Figure 4E) but more detail on the mutations in the binding pocket per se being causal would be helpful.

*Reviewer #2 (Recommendations for the authors):*

I am very positive about this study but do have a couple of requests/suggestions.

1. Since the photoprobe differs significantly from PTX, it would be good to analyze the binding pocket mutations identified in Figure 4 using the direct binding assay of PTX in Figure 5A. Either the mutants or some generic albumin would be a good control for this figure to establish the specificity of ABG.

2. In the proteomic analysis (Figure 2B and elsewhere), where is albumin? Is it surprising that albumin is not a strong hit, especially in light of the earlier literature implicating these proteins as sequesterers of the alkaloids?

---

## [Author Response]

Essential revisions:1) Since the photoprobe differs significantly from PTX, it would be good to analyze the binding pocket mutations identified in Figure 4 using the direct binding assay of PTX in Figure 5A. Either the mutants or some generic albumin would be a good and essential control for this figure to establish the specificity of ABG.

Thank you for this suggestion, we agree these are important controls and have conducted the suggested experiments. Microscale thermophoresis (MST) showing reduced binding of mutant 3 (D383A) and BSA to PTX have been added to Figure 5A and is included in the results and Discussion section as follows:

Lines 213-215 (Results): “Using microscale thermophoresis (MST) we found that wild type OsABG binds PTX with greater affinity than Bovine Serum Albumin (BSA) or OsABG mutant 3 (Figure 5A). OsABG mutant 3, D383A, had similar binding affinity for PTX as BSA (Figure 5A).”

Lines 267-270 (Discussion): “A more detailed mutational scanning would be necessary to fully understand which residues coordinate the binding of different alkaloids, however we see that single point mutations significantly disrupt binding ability (Figure 4F, Figure 5A), suggesting a carefully encoded specificity.”

2) Although the protein gels in Figure 1-2 show clearly the role of ABG, a ~50 kDa protein, it's unclear whether transferrin-like proteins, which are ~80 kDa, may also play a role because the gels show proteins between 39-64 kDa (Figure 1). The gel in Figure 2A is specific to one O. sylvatica and extends this range but the gel does not appear to be labeled accordingly, making it unclear whether other larger proteins could have been detected in addition to ABG. Related to this point, did the proteomic analysis (Figure 2B and elsewhere) reveal any hint of albumin? Is it surprising that albumin is not a strong hit, especially in light of the earlier literature implicating these proteins as sequesterers of the alkaloids? Clarifying these issues would facilitate the interpretation of the results.

We apologize for the ladder labeling issue on Figure 2A, this has been corrected to show that the ladder on the blot goes up to 97 kDa. To add further clarification, we have highlighted where albumin comes out in the proteomic analysis in Figure 2B, and what the individual replicate spectral counts showed for albumin in Figure 2C. We agree that it is surprising given the general abundance of albumin in vertebrate plasma and previous literature. Given the high expression of OsABG that we highlight in Figure 6D and discuss in lines 370-372, it is possible that OsABG is also a very highly abundant protein in *O. sylvatica* plasma. As for its prior implication in alkaloid binding, we do not discount that albumin and other proteins might also be playing a role in alkaloid transport, however our results show that *Os*ABG is a more specific and abundant protein for the alkaloids tested in this study.

The following has been added to the results and discussion to reflect these changes:

Lines 150-152 (Results): “In comparison to ABG, albumin showed high abundance but no competition (Figure 2B,D).”

Lines 259-262 (Discussion): “That albumin is not the primary carrier for PTX in poison frog plasma is surprising given its abundance and previously documented roles in small molecule transport, although it is possible that there are poison frog alkaloids that are transported on albumin instead of ABG.”

3) There is what seems to be a significant size difference between the O. sylvatica bands and bands from the other toxic frog species, namely D. tinctorius and E. tricolor. Could the photoprobe be binding to other non-ABG proteins of different sizes in different frog species? Given that O. sylvatica bands are bright and this species was the only one subject to proteomics quantification, a possible conclusion may be that the ABG toxin sponge is a lineage-specific adaptation of O. sylvatica rather than a common mechanism of toxin sequestration among multiple independent lineages of poison frogs. It would be helpful if the authors could address this observation of their binding data and the hypothesis flowing from that in the manuscript.

We understand that is a possibility that cannot be ruled out without proteomics quantification of *E. tricolor* and *D. tinctorius*. We attempted this quantification, but unfortunately it is inconclusive without a reference genome to map the peptides against. We have added the following to the discussion to reflect the point being made in this comment:

Lines 296-301 (Discussion): “In *D. tinctorius* and *E. tricolor* plasma the photocrosslinking banding pattern and size of the bands was different than that of *O. sylvatica*. We hypothesize that these plasma bands consist of the ABG proteins identified through homology (Figure 4B-C), however because we did not perform proteomics quantification, the possibility remains of other alkaloid binding proteins in *D. tinctorius* and *E. tricolor*, which would suggest that ABG is an evolutionary innovation unique to the O. sylvatica lineage. The identification of *Os*ABG opens up the possibility of further discoveries of ABGs in other toxic species.”

4) Figure 4B-C: Photoprobe binding results in the presence of epi and nicotine appear to be missing for D. tinctorius and those in the presence of PTX and nicotine are missing for D. tricolor. Adding these results would make for a more complete picture of alkaloid binding by ABG in non-O. sylvatica species.

These results have been added to Figure 4B and 4C and the following lines have been added to the results:

Lines 186-189 (Results): “Purified DtABG and EtABG required higher concentrations of protein to see a signal and showed much weaker photoprobe binding, which was competed off by the presence of PTX and DHQ for both DtABG (Figure 4B) and EtABG (Figure 4C).”

5) Using recombinant proteins with mutations at residues forming the binding pocket of O. sylvatica ABG (as inferred from docking simulations), the authors found that all binding pocket mutations disrupted photoprobe binding completely in vitro (L221-222, Figure 4E). However, there is no information presented on non-binding pocket mutations. Mutations outside of the binding pocket would presumably maintain photoprobe binding – barring any indirect structural changes that might disrupt binding pocket interactions with the photoprobe. This result is important for the conclusion that the binding pocket itself is the sole mediator of toxin interactions. The authors do show that one binding pocket mutation (D383A) results in some degree of photoprobe binding (Figure 4E) but more detail on the mutations in the binding pocket per se being causal would be helpful.

While we understand the idea behind making a non-binding pocket mutant, we feel that the results would be difficult to interpret given the possibility of structural changes that affect the binding pocket even if the mutation is made in a distant region. The addition of the mutant 3 (D383A) direct binding data in Figure 5A shows further quantification of that binding pocket mutant, and indicates that the protein is stably folded as we did not see aggregation. To further describe the binding pocket, we also made two additional mutants which we show in Figure 4E and Figure 4F based on the structural predictions and sequence homology, and found that both of these mutations also decreased photoprobe binding yet maintained competition activity by PTX. We have added the following to the results and discussion:

Lines 199-202 (Results): “The single point mutations of D383A (m3), Y36F (m4), and S374A (m5) weakened photoprobe binding significantly, to the point of being nearly undetectable (Figure 4F) in comparison to the wild type protein. All single point mutations retained the ability to compete photoprobe binding with PTX (Figure 4F). ”

Lines 267-270 (Discussion): “A more detailed mutational scanning would be necessary to fully understand which residues coordinate the binding of different alkaloids, however we see that single point mutations significantly disrupt binding ability (Figure 4F, Figure 5A), suggesting a carefully encoded specificity”

We acknowledge that these results do not entirely quell the doubts presented in this comment. The gold standard would be a co-crystal structure and deep mutational scan of the protein, both of which we felt were outside of the scope of the current project. We believe that as is, our results provide strong evidence for the suggested binding pocket that can be followed up on in future work.

Reviewer #2 (Recommendations for the authors):I am very positive about this study but do have a couple of requests/suggestions.1. Since the photoprobe differs significantly from PTX, it would be good to analyze the binding pocket mutations identified in Figure 4 using the direct binding assay of PTX in Figure 5A. Either the mutants or some generic albumin would be a good control for this figure to establish the specificity of ABG.2. In the proteomic analysis (Figure 2B and elsewhere), where is albumin? Is it surprising that albumin is not a strong hit, especially in light of the earlier literature implicating these proteins as sequesterers of the alkaloids?

Thank you for the positive review of our work and the helpful suggestions! We have addressed both of your comments in the “essential revisions” section above.